# Component Design of Environmentally Friendly High-Temperature Resistance Coating for Oriented Silicon Steel and Effects on Anti-Corrosion Property

Ying Liu [1,2], Lin Wu [1,2,3], Ao Chen [1], Chang Xu [1], Xiaoyu Yang [1], Yilai Zhou [1], Zhiyuan Liao [1], Baoguo Zhang [1], Ya Hu [1,2,3,*] and Hailiang Fang [2,3,*]

1   Key Laboratory of Hubei Province for Coal Conversion and New Carbon Materials, School of Chemistry and Chemical Engineering, Wuhan University of Science and Technology, Wuhan 430081, China; liuying-9325@wust.edu.cn (Y.L.); wulin@wust.edu.cn (L.W.); chen_ao@wust.edu.cn (A.C.); xuchang@wust.edu.cn (C.X.); zhlyxy@wust.edu.cn (X.Y.); zhouyilai@wust.edu.cn (Y.Z.); liaozhiyuan@wust.edu.cn (Z.L.); zbg@wust.edu.cn (B.Z.)
2   Hubei Engineering Technology Research Center of Marine Materials and Service Safety, Wuhan University of Science and Technology, Wuhan 430081, China
3   Academy of Green Manufacturing Engineering, Wuhan University of Science and Technology, Wuhan 430081, China
*   Correspondence: huya@wust.edu.cn (Y.H.); simbaqiankun@163.com (H.F.)

**Abstract:** Oriented silicon steel is vital for power transformer cores, while the high-temperature annealing process limits the industrialization of environmentally friendly coatings on the surface. In this paper, the high-temperature binders $Al(H_2PO_4)_3$ solution and silica sol were introduced innovatively. They condensed into macromolecular polymer chains, network structures and $SiO_2$ particles at high temperatures, providing high-temperature stability and adhesion. The influence of types of silica sol, additives and functional fillers on the corrosion resistance of the coating was studied. The prepared environmentally friendly inorganic insulating coating for oriented silicon steel has excellent corrosion resistance after curing at 475 °C and annealing at 800 °C, which was matched with the currently rolling process of oriented silicon steel. The salt-spray resistance can last for more than 24 h and up to 72 h.

**Keywords:** environmentally friendly coating; inorganic coating; oriented silicon steel; high-temperature resistance; corrosion resistance



## 1. Introduction

Oriented silicon steel is widely used as a power transformer core, while the manufacturing process is complex and technically difficult. Oriented silicon steel is a "sandwich" structure composed of a silicon-steel substrate, a magnesium silicate ($MgSiO_3$) mezzanine and an insulating coating [1,2]. The insulating coating on the surface is vital and needs to have good insulation, corrosion resistance, adhesion and heat resistance, which can delay the occurrence of corrosion to prolong the service life of the transformer, avoid short circuits and reduce the eddy current loss of electrical steel sheets [3,4]. At present, the commercial insulating coating for oriented silicon steel is still chromium-containing. Although it has excellent performance, it is harmful to humans and the environment. The development of a chromium-free insulating coating is critical. However, most of the chromium-free insulating coatings for oriented silicon steel cannot simultaneously achieve the technical effects of chromium-containing coatings or withstand the high temperatures required for oriented silicon-steel rolling, since the organic components are easy to decompose when heated. According to the current oriented silicon-steel-rolling process, the insulating coating needs to be cured at a low temperature (≤500 °C) and then annealed at a high temperature (≥800 °C) [5–7], and chromium-free coatings with good performance such as

those reported by D. Zhang et al. [8] have performance degradation after annealing for the pyrolysis of the acrylic resin included. Therefore, making the coating have both good performance and high-temperature resistance is the focus of the research on chromium-free insulating coatings for oriented silicon steel.

Common insulating coatings are mainly organic coatings, semi-inorganic coatings, and inorganic coatings, which will have different effects on the properties of oriented silicon steel. The organic coating uses organic resin to form the film, which has the characteristics of insulation, good film-forming properties and good impact resistance, but its hardness and weldability are poor. Furthermore, it is easy to shrink and deform after being heated and releases pollution gas at the same time [9]. The semi-inorganic coating is currently the most widely used and is formed by mixing organic resin with inorganic solutions. That is, organic resin and inorganic solution are mixed uniformly and then coated on the surface of electrical steel for curing. The semi-inorganic coating has good adhesion and shear punching property [10–12]. Usually, the organic resin could be polyamide, polypropylene, polyethylene, benzo-diamino triazine, water-based epoxy resin or acrylic resin and the inorganic component of the representative semi-inorganic coating for oriented silicon steel is chromate. In some reports about chromium-free coatings, the inorganic component could be phosphate, rare-earth salts, borate and nanomaterials such as nanosilicon dioxide or graphene [13–15]. The semi-inorganic coating has good corrosion resistance, but due to the poor high-temperature resistance of the organic part in the coating, the coating will significantly lose weight when the temperature exceeds 400 °C, and the rate can even reach 90%. This will greatly reduce the performance [16,17]. At present, inorganic coatings are mainly divided into phosphate-system coatings, electrodeposition coatings, coatings prepared by the gel–sol method and ceramic coatings [18–22]. Inorganic coatings have the advantages of high-temperature resistance, good welding performance, small thermal expansion coefficient and large tensile stress. However, their hardness is too large, and their punching shear performance and corrosion resistance are poor. Aluminum dihydrogen phosphate ($Al(H_2PO_4)_3$) and silica sol are commonly used high-temperature binders, which can be dehydrated at high temperature into macromolecular polymer chains and network structures, providing high-temperature stability and adhesion [23]. J.Y. Jia et al. [24] and J.B. Liu et al. [25] pointed out that aluminum dihydrogen phosphate ($Al(H_2PO_4)_3$) can be converted into aluminum dihydrogen tripolyphosphate ($AlH_2P_3O_{10}\cdot 2H_2O$), aluminum phosphate ($AlPO_4$), or $Al(PO_3)_3$ at different temperature stages. X.B. Wang et al. [26] found that the silica sol in the white corundum–silica sol coating exists in the form of crystalline silicon dioxide ($SiO_2$) and $Al_9Fe_2Si_2$ formed by the reaction with the coating at 600 °C. S. Yao et al. [27] believed that silica sol formed quartz-type $SiO_2$ at a temperature higher than 1300 °C. Therefore, $Al(H_2PO_4)_3$ and silica sol can be used in oriented silicon-steel insulation coatings and other water-based coating systems that have certain requirements for temperature resistance. The objective of this work is to design a chromium-free insulating coating with high-temperature binder $Al(H_2PO_4)_3$ and silica sol to keep high-temperature resistance and good corrosion resistance simultaneously and investigate the impact of component characteristics, including the particle size and acidity of the silica sol, different types of additives and functional fillers, on corrosion resistance.

## 2. Materials and Methods

### 2.1. Materials

The silica sol was provided by Shandong Yinfeng Nano New Materials Co., Ltd., Jinan, Shangdong, China. The content of $SiO_2$ is 30 wt%. The $Al(H_2PO_4)_3$ solution was purchased from Zhengzhou Yucai Phosphate Factory, Zhengzhou, Henan, China. The content of aluminum oxide ($Al_2O_3$) is 7.5–8.5 wt%, and that of phosphorus pentoxide ($P_2O_5$) is 32–34 wt%. The magnesium oxide (MgO, AR) and sodium chloride (NaCl, AR) were purchased from Sinopharm Chemical Reagent Co., Ltd., Shanghai, China. The deionized water was homemade.

## 2.2. Coatings Preparation

According to the formula, different silica sols are mixed, and $Al(H_2PO_4)_3$ solution, and liquid additives, and deionized water are sequentially added to the mixed silica sol and stirred uniformly to obtain chromium-free inorganic insulating coatings. When solid additives are involved in the formulation, they are dissolved in $Al(H_2PO_4)_3$ solution or deionized water before being added to the coating. The chromium-free insulating coating is suitable for oriented silicon-steel sheets. The oriented silicon-steel sheets were washed with a soft brush using ethanol, 5 vol% sulfuric acid ($H_2SO_4$) and running water in sequence. Then, the sheets were applied using a wire-wound rod coater, and the coating was cured at 475 °C and 800 °C for 10 s and 40 s, respectively.

## 2.3. Characterization

The heat resistance of the coating was investigated by thermogravimetric analysis (TGA). The chromium-free inorganic insulating coating was cured at 100 °C, ground, and was analyzed by differential thermal analysis (STA449, NETZSCH, Selb, Germany). The temperature was increased from 20 °C to 900 °C by a temperature-rising rate of 10 °C/min in a nitrogen atmosphere. The phase structure was investigated by X-ray diffraction (XRD, Cu kα, SmartLab SE, Rigaku Corporation, Tokyo, Japan). The micromorphology was observed by scanning electron microscope (SEM, Philps-XL30 TMP, Philps, Amsterdam, Netherlands). The corrosion resistance of the coating was investigated by the neutral salt-spray test (NSS) and the electrochemical test. According to the experimental method in GB/T 10125–2012, the neutral NaCl aqueous solution was used and a continuous spray was applied to accelerate corrosion. The concentration of NaCl solution used is 45–55 g/L. The ambient temperature is maintained at 33–37 °C in the experiment. The sedimentation volume of salt spray is between 1~2 mL/h. The electrochemical test was carried out with the electrochemical workstation (CH 760E, Shanghai Chenhua Instrument Co., Ltd., Shanghai, China). Electrochemical Impedance Spectroscopy (EIS) and Tafel polarization curve test were applied using a conventional three-electrode cell (Ag/AgCl as a reference electrode, a sample plate as a working electrode, and a platinum electrode as an auxiliary electrode). The electrolyte solution is the 3.5 wt% NaCl solution. Measurements at the open circuit were carried out at an AC frequency range from 100 kHz to 0.01 Hz, with an amplitude of 5 mV. The Tafel polarization curve was adopted in the voltage range of −0.5–0.5 V with the open-circuit potential as the midpoint, and the scan rate was 0.01 V/s.

## 3. Results and Discussion

We selected $Al(H_2PO_4)_3$ and silica sol as the main film-forming materials and added metal oxides and some additives to improve the leveling and stability of the coating, then obtained an inorganic chromium-free coating. In order to check the heat resistance of the coating, the TGA analysis was carried out with a semi-inorganic coating we made [12] as a contrast. The analysis was carried out in a nitrogen atmosphere which was the same condition in the rolling process. The test temperature was up to 900 °C, which is higher than the requirements of the rolling process. The results are shown in Figure 1.

Figure 1a is the TGA of semi-inorganic coating, including the thermogravimetric (TG) curve, differential thermal gravity (DTG) curve and differential scanning calorimetry (DSC) curve. This coating contains epoxy resin. It has excellent corrosion resistance and the corrosion area is 10% after NSS for 120 h. There are two decreasing sections and three peaks representing extreme points of mass loss in the DTG curve at 78.9 °C, 426.7 °C and 470.6 °C. The first descending section in the TG curve and peak at 78.9 °C indicate that there is remaining free water and crystal water in the coating powder and the water is evaporated with the heat process. The second descending section in the TG curve and peaks at 426.7 °C and 470.6 °C indicate that the epoxy resin and the organic curing agent are decomposed. The remaining mass is 60.78% at 849.8 °C. The TGA of semi-inorganic insulating coating shows that the appropriate temperature for curing should be less than 400 °C. The actual curing temperature for the semi-inorganic coating is 300 °C, and the

corrosion area is more than 50% after NSS for 1h when cured at 800 °C. Figure 1b is the TGA of inorganic coating. There is an increasing section and a decreasing section in the TG curve. Corresponding to the sections, there is an upward peak at 27.3 °C representing the extreme point of mass addition and a downward peak at 42.9 °C representing the extreme point of mass loss. In our experiment, there was an interesting phenomenon that the cured coating would absorb moisture and become moist and sticky when stored at room temperature without being annealed at 800 °C. The increasing section in the TG curve and upward peak at 27.3 °C matched with the phenomenon. However, with the increase in temperature, the remaining free water and crystal water began to be evaporated, which matched with the decreasing section in the TG curve and downward peak at 42.9 °C. The remaining mass was 88.02% at 899.8 °C. It is worth noting that the DSC curve reveals that the coating is a continuously endothermic process, while the mass is almost constant. That indicates that the phase structure of the coating changes during the heat process.

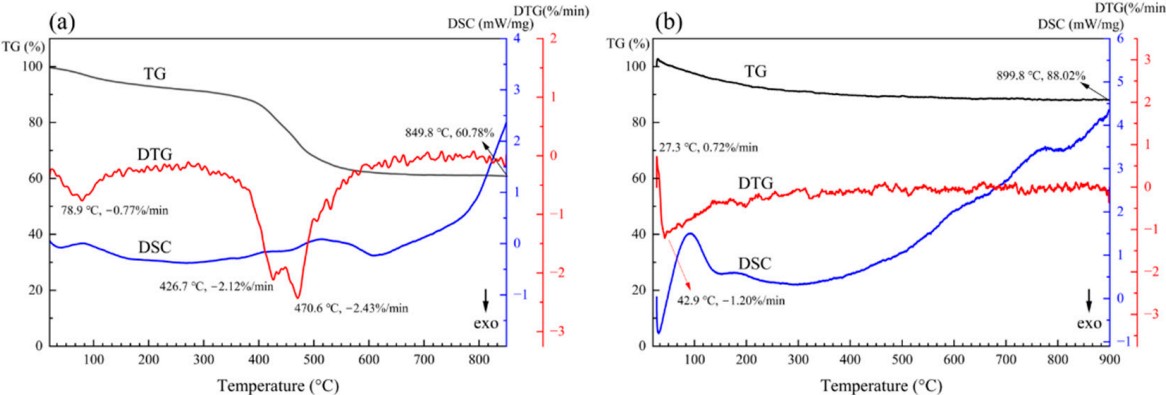

**Figure 1.** The TGA results of the coatings: (**a**) chromium-free semi-inorganic coating and (**b**) chromium-free inorganic coating.

To reveal the structural changes of the coating during the heat process, the coating was dried at 60 °C and ground before high-temperature heat treatment. Then, the samples were taken at two stages of treatment and XRD tests were carried out. Figure 2 was the XRD patterns of samples with different heat-treatment temperatures. It can be found that in different heat-treatment stages, the peak shape changed significantly while the positions were the same. Specifically, there was broad and dispersive peak located at 15°–35° in the pattern representing the amorphous $AlPO_4$ and $SiO_2$ when dried at 60 °C. Since the peak positions of $AlPO_4$ (ICDD/JCPDS 11-0500) and $SiO_2$ (ICDD/JCPDS 27-0605) were close, it was difficult to directly separate the two phases. After the heat treatment at 475 °C, a sharp peak could be observed near 21°, which represents the $AlPO_4$ (111) and the $SiO_2$ (111). In addition, after heat treatment at 800 °C, the intensity of the peak around 21° was further enhanced in the XRD pattern, which indicates that the content of $AlPO_4$ and $SiO_2$ transforming into crystalline state increased. Furthermore, a sharp peak appeared at around 35° corresponding to $AlPO_4$ (220) and $SiO_2$ (220). Comparing the XRD patterns of the coatings at different heat-treatment stages, it can be concluded that with the increase in the heat-treatment temperature, the composition of the coating does not change significantly, and the content of crystalline $AlPO_4$ and $SiO_2$ increases. Since the failure of the coating in the heat process is usually related to the burnout at high temperature, combining the XRD and TGA results can further prove that the prepared coating has high-temperature resistance.

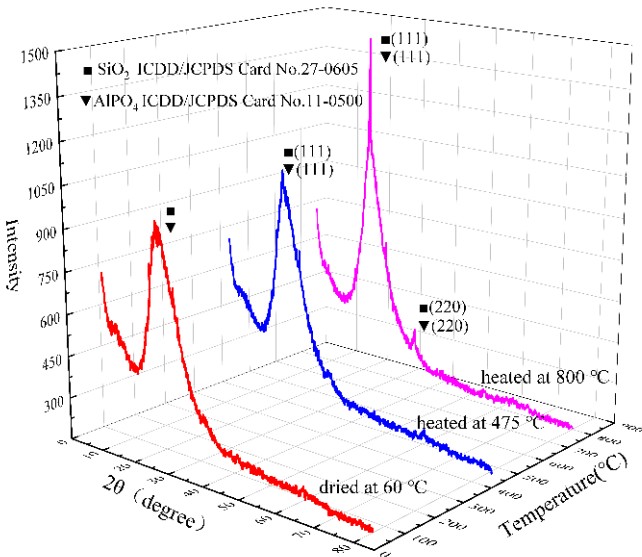

**Figure 2.** The XRD patterns of the coatings after different heat treatment.

The coating can be cured in situ by combining with the rolling process of oriented silicon steel, which means there is no need to adjust the parameters in the rolling process as shown in the Figure 3.

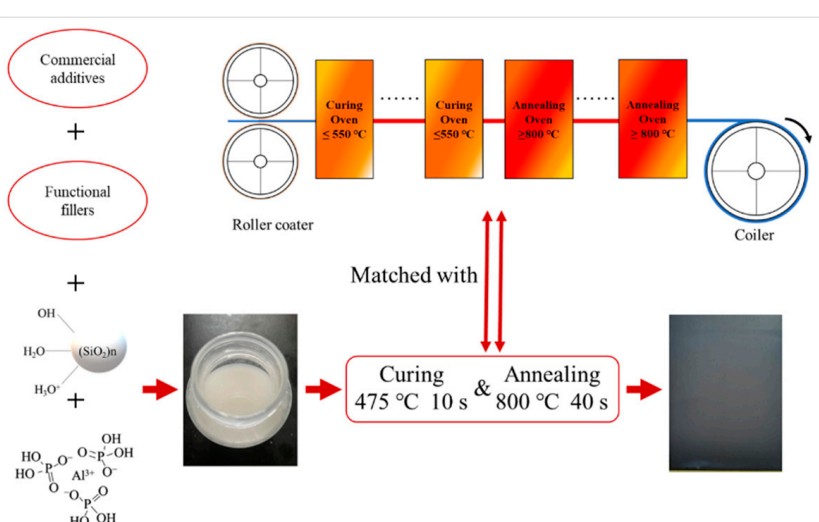

**Figure 3.** The schematic diagram of in situ curing process.

According the TGA analysis result, the coating was cured at 475 °C for 10 s and sintered at 800 °C for 40 s, which is the heat treatment currently used for oriented silicon-steel manufacture. The surface of coating was bright gray. There were no visible cracks or pores when observed by naked eyes. Since the coating achieves corrosion resistance through shielding, that is, it prevents the corrosive medium from contacting the metal base, the coating morphology has an important impact on the corrosion resistance of the coating. We observed the morphological changes of the oriented silicon-steel sheet before and after coated by SEM, as shown in Figure 4.

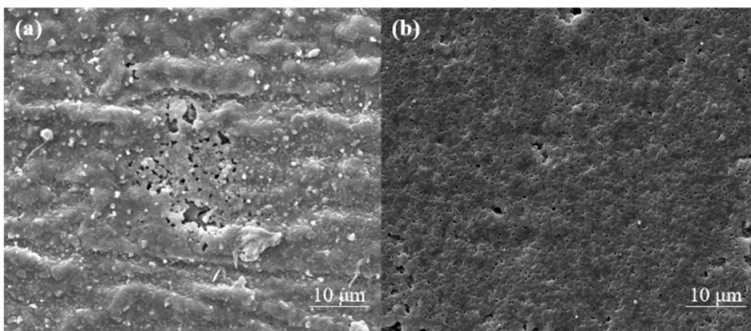

**Figure 4.** The micromorphology observed by SEM: (**a**) blank plate and (**b**) coated plate.

Figure 4 is the micromorphology observed by SEM when magnified 5000 times. The blank plate is $MgSiO_3$ mezzanine (Figure 4a). The $MgSiO_3$ mezzanine is composed of agglomeration and accumulation of particles and partially damaged. The coated plate is smoother and has fewer defects (Figure 4b). The morphology of the coating also showed a grainy appearance, which indicates that the coating may be silica particles polymerized and bonded by $Al(H_2PO_4)_3$ to form a shielding film. The film is flatter without obvious cracks and the pores are smaller without the exposed metal substrate. This is beneficial to improve the corrosion resistance of the coating.

The NSS test was carried out for a further study on corrosion resistance.

Figure 5 shows the NSS results of coated boards and blank boards. The results show that although the corrosion-resistance time has been reduced, the adjusted chromium-free coating still has good corrosion resistance. The corrosion area after 48 h of neutral salt-spray teat is less than 5%, while the corrosion area of the blank board after 2 h exceeds 90%. This is lower than the current industry requirement that the corrosion area is less than 5% in 5 h [28].

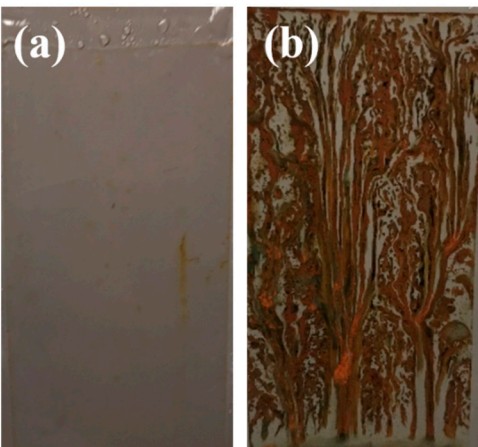

**Figure 5.** The NSS results: (**a**) coated plate for 48 h and (**b**) blank plate for 2 h.

Figure 6 shows the EIS results of the chromium-free insulating coating for oriented silicon steel. Figure 6a is the Nyquist plot of the coated plate and the blank plate. As shown in the figure, the impedance radius of the coated plate is much larger than that of the blank plate, indicating that the coating effectively improves the corrosion resistance. Figure 6b shows the Bode plots of the coated plate and the blank plate. The impedance modulus value in the low-frequency area is effective in evaluating the corrosion resistance of the sample [29]. The low-frequency impedance modulus value of the coated board is two orders of magnitude higher than that of the blank board, which also indicates that the coating can effectively provide corrosion resistance.

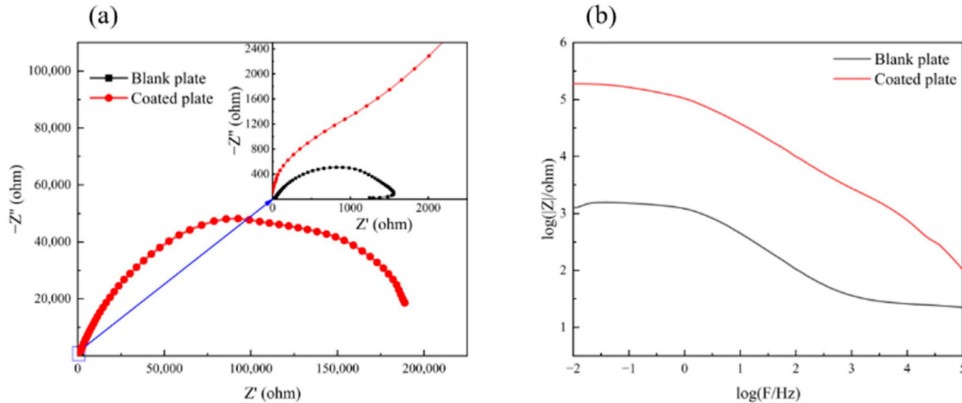

**Figure 6.** The EIS test results: (**a**) Nyquist plots and (**b**) Bode plots.

Figure 7 is the test result of the Tafel polarization-curve test carried out on blank plates and coated plates. Table 1 shows the fitting parameters of the Tafel polarization curve. According to the fitting results, compared with the blank plate, the corrosion potential (Ecorr) of the coated plate is positively shifted, and the corrosion current density (Icorr) is reduced by about two orders of magnitude. This indicates that the coating can provide better corrosion resistance in terms of hindering the occurrence of corrosion and delaying the rate of corrosion occurrence. Combining the NSS and electrochemical test results, the chromium-free insulating coating of oriented silicon steel can effectively retard the occurrence and progress of corrosion and improve the corrosion resistance of oriented silicon steel.

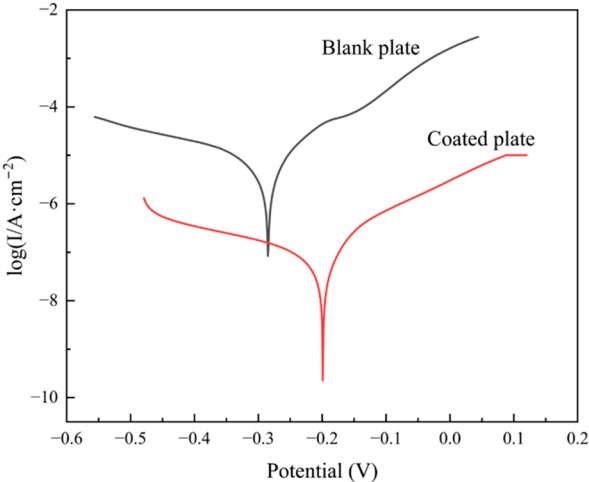

**Figure 7.** The Tafel polarization curves of the blank plate and coated plate.

**Table 1.** Tafel polarization parameters of the blank plate and coated plate.

| Sample | Ecorr (V) | Icorr ($\mu A \cdot cm^{-2}$) |
|---|---|---|
| Blank plate | −0.3013 | 9.8175 |
| Coated Plate | −0.2031 | 0.0890 |

The above experimental results show that we have successfully designed a chromium-free inorganic insulating coating for oriented silicon steel, which can be cured in situ and has both high-temperature resistance and corrosion resistance. In the coating-formulation design process, the experiments show that the component characteristics have a direct impact on the morphology and corrosion resistance of the coating. These component

characteristics include the particle size and acidity of silica sol, metal-ion types and addition methods, and additives.

Through the above crystal-phase state and morphology analysis, it can be speculated that the structure of the coating is the combination of $AlPO_4$ which provides structural stability through a three-dimensional network structure [23] and stacked $SiO_2$ particles. This structure has a similar shielding effect as organic or semi-inorganic coatings; that is, the coating protects the substrate by keeping it from the penetration of corrosive. When the organic or semi-inorganic coating is pyrolyzed and loses weight after high-temperature annealing, $CO_2$ and $H_2O$ as the decomposition products will escape from the coating and leave pores and cracks. The compactness of the coating is reduced and corrosion will be easier. The inorganic coating that we designed still has a complete structure to realize effective coverage after high-temperature annealing, which guarantees the corrosion resistance of the coating. For the coating based on the shielding effect, making the path of the corrosive more tortuous will be an effective way to improve corrosion resistance, such as the addition of functional fillers graphene oxide [30]. For our coating, we believe that changing the particle size of the silica sol will be such a way.

Figure 8 demonstrates the topographic characteristics of the coating consisting of $SiO_2$ particles with different size. In the figure, the film thickness is fixed. This is because the thinner the insulating coating is, the better. The coating thickness on the surface of oriented silicon steel is usually 0.5–2.5 µm. Otherwise, the thick film would cause the decrease in metal proportion and influence the application of iron core, which consists of oriented silicon-steel sheets stacked up. In the figure, it is also observed that some $SiO_2$ particles are above the $AlPO_4$ net and some particles are under the net. This is based on the SEM and XRD analysis results, and we believe that $Al(H_2PO_4)_3$ transforms into $AlPO_4$ constituting a three-dimensional network structure, and the silica sol transforms into $SiO_2$ particles interspersing in the net and stacking on the substrate. The length of the corrosive-diffusion path in the coating consisting of normal-size $SiO_2$ particles is $L_1$. When the thickness is fixed, increasing the size would reduce the amount of $SiO_2$ or even the number of layers and the intergranular pore width would increase. The length of the corrosive diffusion path in such coating is $L_2$. On the contrary, decreasing the size would increase the amount of $SiO_2$, reduce the intergranular pore width and change the length of the corrosive-diffusion path to $L_3$. When combining particles of different sizes, the small particles would fill the blanks among big particles and the path would be more complex and irregular. The length in such coatings is $L_4$. It can be found that $L_3$ is greater than $L_1$ and $L_1$ is greater than $L_2$. $L_4$ is depended on the difference between the two particle sizes, and the larger the difference, the more complex the path will be; that is, the increase in tortuosity. The increase in the length and tortuosity of the corrosion-penetration path can delay or even prevent the corrosion, and realize the increase in the corrosion resistance of the coating. Based on this inference, we selected silica sol with different particle sizes to prepare a chromium-free inorganic coating according to the formula shown in Table 2, and observed the morphology of the coating.

**Table 2.** Coatings with silica sols of different particle sizes.

| Sample | $Al(H_2PO_4)_3$ (g) | Silica Sol (g) | | | | | Deionized Water (g) |
|---|---|---|---|---|---|---|---|
| | | 8 nm | 10 nm | 15 nm | 30 nm | 60 nm | |
| Si10-15 | 35 | - | 20 | 20 | - | - | 25 |
| Si10-30 | 35 | - | 20 | - | 20 | - | 25 |
| Si10-60 | 35 | - | 20 | - | - | 20 | 25 |
| Si8-10 | 35 | 20 | 20 | - | - | - | 25 |
| Si-All | 35 | 10 | - | 10 | 10 | 10 | 25 |

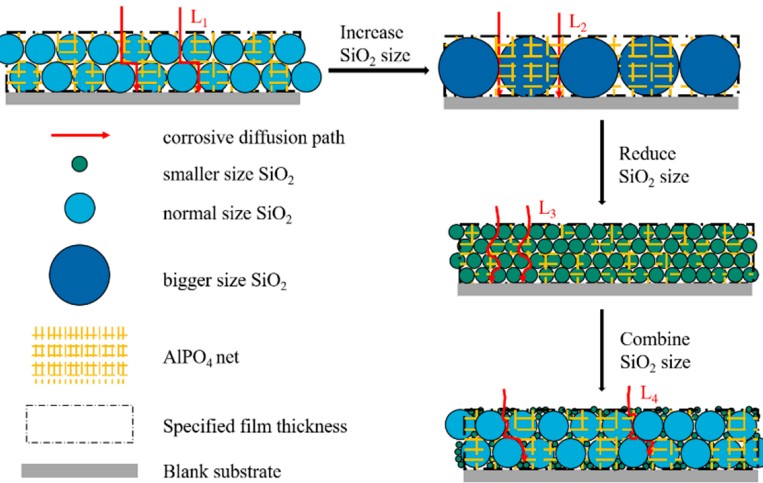

**Figure 8.** Scheme of the effect of silica sol particle size on the corrosion resistance.

The experimental results show that all coatings can form a film to cover the surface of the oriented silicon-steel sheet after heat treatment at 475 °C for 10 s and 800 °C for 40 s, but samples coated with different coatings have different morphological characteristics. Among them, Si10-15 has higher requirements for the amount used. When the amount is too small, the surface will be reddish, which means that the coating is so thin that defects and oxidation come out during the film formation. The surface will whiten where the coating accumulates, even if the amount of coating is slightly excessive. Unlike Si10-15, the surface of Si10-30 or Si10-60 is uneven and there is always a whiter place regardless of the amount. Even more, some area of the Si10-60 is chalked. The edge of the Si8-10 shrinks. In addition, when increasing the amount, there is no significant improvement, but some unevenly whiter areas appear on the surface due to the partial accumulation of the coating. Shrinkage holes and shrinkage edges appear easily on the surface of Si-All, and the color of the thicker part is darker and makes the surface look uneven. According to those results, it is harmful to the leveling characteristics of the coating when too many particle-size categories of silica sol are used. When the particle size is too large, the surface is more likely to be whitened. If the particle size is too small, the surface will have a higher correlation with the coating consumption. This may be caused by the expected independent particle structure not being formed at the specified temperature when the silica sol particle size is too large or the coating accumulates unevenly. Instead, the silica is accumulated. Since the morphology of the samples did not meet the requirements, the NSS was not carried out. However, the overall results indicate that the coating formed by $Al(H_2PO_4)_3$ and silica sol can be used for oriented silicon steel. Finally, 8 nm silica sol and 15 nm silica sol are used for combination.

In the process of study about the particle size of silica sol, gelation occurred after a period when mixing alkaline silica sol and $Al(H_2PO_4)_3$ solution. Considering that the $Al(H_2PO_4)_3$ solution is acidic while the silica sol is alkaline, it may have been caused by the acid-base neutralization reaction changing the charge distribution and hydration radius of the colloidal particles, which affected the stability of silica sol [31]. To weaken the structural change of silica sol, we selected silica sol with different acidity and different particle size, prepared chromium-free inorganic coatings according to the formula shown in Table 3 and observed the coating morphology after curing (Figure 9). The coating with good morphology was tested by NSS, and the experimental results are shown in Figure 10.

**Table 3.** Coatings with silica sols of different particle sizes and acid base.

| Sample | Al(H$_2$PO$_4$)$_3$ (g) | Silica Sol (g) | | | | | | Deionized Water (g) | Additive (g) |
| | | Alkaline | | Acidic | | | Neutral | | |
| | | 8 nm | 15 nm | 8 nm | 15 nm | 30 nm | | | |
|---|---|---|---|---|---|---|---|---|---|
| Al8-Ac15 | 35 | 20 | - | - | 20 | - | - | 25 | - |
| Al15-Ac8 | 35 | - | 20 | 20 | - | - | - | 25 | - |
| Ac8 | 35 | - | - | 40 | - | - | - | 25 | - |
| Ac8-Ac15 | 35 | - | - | 20 | 20 | - | - | 25 | - |
| Ac15-Ac30 | 35 | - | - | - | 20 | 20 | - | 24.3 | 0.7 |
| Ac30 | 35 | - | - | - | - | 40 | - | 24 | 1 |
| Neu | 35 | - | - | - | - | - | 40 | 24.5 | 5 |

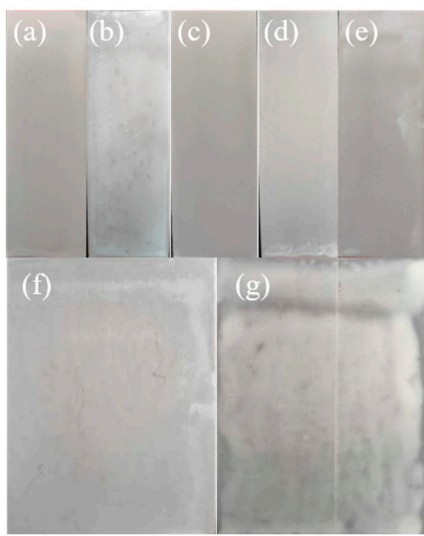

**Figure 9.** The morphology of coatings with silica sols of different particle sizes and acid base: (**a**) Al8-Ac15, (**b**) Al15-Ac8, (**c**) Ac8, (**d**) Ac8-Ac15, (**e**) Ac15-Ac30, (**f**) Ac30 and (**g**) Neu.

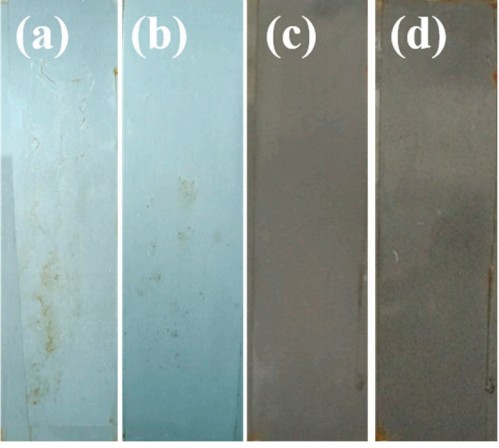

**Figure 10.** The morphology of coatings after NSS: (**a**) Al8-Ac15 after 72 h, (**b**) Ac8 after 72 h, (**c**) Ac8-Ac15 after 72 h and (**d**) Ac8-Ac15 after 96 h.

Experiments show that although the acid base of the silica sol has changed, there is not much difference in the stability of the coatings, and it is more difficult to completely restore the fluidity after gelling when containing 8 nm acidic silica sol. However, the results show that the acidity of silica sol will affect the morphology of the coating. As

shown in Figure 8, Al8-Ac15 (Figure 9a), Ac8 (Figure 9c) and Ac8-Ac15 (Figure 9d) have a good appearance. There are shrinkage holes and chalked areas on Al15-Ac8 (Figure 9b) and the Neu (Figure 9g). The surface of Ac15-Ac30 (Figure 9e) is slightly uneven. Ac30 (Figure 9f) is uniform but whitened. Therefore, although changing the acidity of the silica sol cannot improve the stability of the coating, it will affect the morphology of the sample. Comprehensively considering the influence of the acidity and particle size of the silica sol, the use of 8 nm alkaline silica sol and 15 nm acidic silica sol can make a better plate morphology. The NSS of the Al8-Ac15, Ac8 and Ac8-Ac15 samples were carried out. By comparing their corrosion area after 72 h (Figure 10), it can be found that the corrosion resistance of Ac8-Ac15 is the best, and that of Al8-Ac15 is the worst. In addition, the corrosion area of Ac8-Ac15 after 96 h is still less than 5% (Figure 10d), which shows that the chromium-free insulating coating can provide the oriented silicon steel excellent corrosion resistance by adjusting the formula, but the stability and wetting of the coating need to be improved.

Although the corrosion resistance of the coating is improved by adjusting the acidity and particle size of silica sol, the morphology of the coating is unstable. When the operators of the wire-wound rod coater are different, the morphology of the plate is greatly different. It indicates that the leveling and wettability of the coating still need to be improved. In the coatings industry, there are application examples of commercial additives to improve leveling, stability and wettability [32]. Therefore, we select relevant additives to prepare coatings according to the problems presented, and the types of additives used are shown in Table 4. The experimental results show that the additives have a great influence on the stability of the coating, and the phenomena during preparation and storage are listed in Table 5.

**Table 4.** Coatings with different additives.

| Sample | Al(H$_2$PO$_4$)$_3$ (g) | Acidic Silica Sol (g) | | Deionized Water (g) | Additives (g) |
|--------|------------------------|-----------------------|--------|---------------------|---------------|
| | | 8 nm | 15 nm | | |
| Ad-1 | 35 | 20 | 20 | 25 | ① 0.5 |
| Ad-2 | 35 | 20 | 20 | 25 | ① 0.5 ② 0.1 |
| Ad-3 | 35 | 20 | 20 | 25 | ① 0.5 ② 0.1 ③ 0.3 |
| Ad-4 | 35 | 20 | 20 | 25 | ① 0.5 ② 0.1 ③ 0.3 ④ 0.3 |
| Ad-5 | 35 | 20 | 20 | 25 | ① 0.5 ② 0.1 ③ 0.3 ④ 0.3 ⑤ 0.2 |
| Ad-6 | 35 | 20 | 20 | 25 | ① 0.5 ② 0.1 ③ 0.3 ④ 0.3 ⑤ 0.2 ⑥ 0.2 |
| Ad-7 | 35 | 20 | 20 | 25 | ① 0.5 ② 0.1 ③ 0.3 ④ 0.3 ⑤ 0.2 ⑥ 0.2 ⑦ 0.5 |
| Ad-8 | 35 | 20 | 20 | 25 | ① 0.5 ② 0.1 ③ 0.3 ⑦ 0.5 |

① acidic dispersants, ② nanoneedle-silicon defoamer, ③ bactericide, ④ wetting agent, ⑤ organic defoamer, ⑥ leveling agent, ⑦ MgO.

**Table 5.** The stability of coating including different additives.

| Sample | Ad-1 | Ad-2 | Ad-3 | Ad-4 | Ad-5 | Ad-6 | Ad-7 | Ad-8 |
|--------|------|------|------|------|------|------|------|------|
| Preparation [1] | − | − | − | + | + | + | + | − |
| Stability [2] | Normal | Normal | Normal | Gelation | Gelation | Gelation | Gelation | Normal |

[1] Noted as "−" when great viscosity change is not observed during the preparation process and "+" otherwise.
[2] Stored at room temperature for 24 h.

Among them, the viscosity of Ad-1, Ad-2, Ad-3 and Ad-8 did not change, while Ad-4, Ad-5, Ad-6 and Ad-7 all appeared to thicken. After 24 h of storage at room temperature, Ad-1, Ad-2, Ad-3 and Ad-8 had normal fluidity, while flocculation and sedimentation came out in Ad-4, Ad-5, Ad-6 and Ad-7. Therefore, the wetting agent causes thickening and shortens the stabilization time. In addition, Ad-2, Ad-3 and Ad-9 showed white

precipitation due to the nanoneedle-silicon defoamer. The nanoneedle-silicon defoamer is silicon with the shape of a needle and its particle size is 5 nm. The nanoneedle-silicon defoamer can puncture the bubbles in coating with its sharp ends, but it will form white precipitate quickly since it is insoluble in water and cannot react with other components in the coating. We prepared oriented silicon-steel sheets with coating that were stable after storage, and the morphological characteristics are shown in Figure 11.

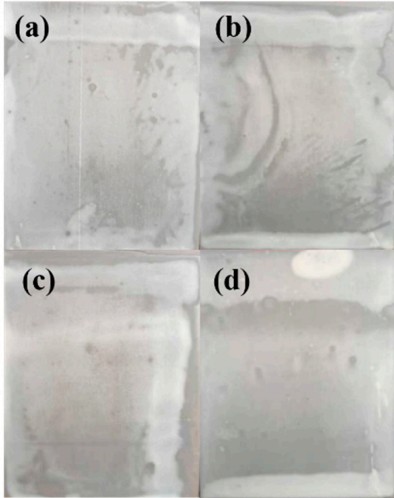

**Figure 11.** The morphology of coatings: (**a**) Ad-1, (**b**) Ad-2, (**c**) Ad-3 and (**d**) Ad-8.

Ad-1, Ad-2, Ad-3 and Ad-8 are used for oriented silicon-steel sheets. In Figure 10, all the surfaces of the sheets are white and powdery, not grey and bright as it supposed to be, so the acidic dispersant is not suitable for a chromium-free insulating coating system. In addition, the water-dispersion stability of the needle-silicon defoamer is limited; it will deposit with the increase in storage time. However, the effect of MgO on the performance of the coating is covered by the effects of other additives, which need further study.

There are reports that MgO can improve the curing speed of $Al(H_2PO_4)_3$ and improve the compactness of the formed coating. The use of MgO as a curing agent can change the curing speed of $Al(H_2PO_4)_3$ and the rate of generation and escape of water vapor during polycondensation, so the coating structure is optimized [33]. Although the additives concealed the role of MgO, we still chose to use MgO as a filler to improve the performance of the coating. However, MgO is a solid powder with low solubility and cannot be directly added to the coating to form a stable system. Therefore, we must design some methods to add MgO.

Taking the Ac8-Ac15 coating system as an example, the methods for adding MgO to the chromium-free insulation coating are shown in Table 6.

As the result, MgO was insoluble and became hard particles in Method A. In Method B, the mixture would be uniform after being stirred for 10 min. After silica sol was added, it became turbid. There was no improvement after continually stirring for 30 min. The paint was half-transparent after filtered, with a loss of 12 g. In Method C, there was no particle generating when adding $Al(H_2PO_4)_3$ into the mixture of MgO and deionized water. However, it became milky white after adding silica sol. In Method D, when adding MgO and stirring for a few minutes, the mixture became jelly-like. After adding $Al(H_2PO_4)_3$, the mixture returned to flow with some insoluble matter that may be MgO. If the stirring time was only 2–3 min after adding MgO and $Al(H_2PO_4)_3$ was immediately added, there was no intermediate state but MgO would remain even after being stirred for 30 min. In Method E, the mixture quickly became gel after adding MgO. In Method F, the mixture became opaque when adding MgO and became transparent as the MgO was dissolved, but it became milky white with the addition of silica sol and the viscosity became larger. Furthermore, the viscosity increased after continuous stirring for 30 min. Method G had

the same phenomenon as Method F at the first step, but there was no visible change in viscosity and transparency when adding silica sol. In Method H, the paint was always stable and this operation was simple, which was conducive to processing and production. Therefore, the appropriate way to add MgO is to dissolve in part of $Al(H_2PO_4)_3$, mix with the remaining $Al(H_2PO_4)_3$, and then use it in chromium-free insulation coatings. There are no obvious defects in the morphology of sheets coated with the paint prepared by this method. The corrosion area was less than 5% after NSS for 72 h, as shown in Figure 12b.

**Table 6.** Different methods to add MgO.

| Reagent / Step Order Method | 1 | 2 | 3 | 4 | 5 |
|---|---|---|---|---|---|
| A | $Al(H_2PO_4)_3$ | silica sol | $H_2O$ | MgO | |
| B | $Al(H_2PO_4)_3$ | $H_2O$ | MgO | silica sol | |
| C | $H_2O$ | MgO | $Al(H_2PO_4)_3$ | silica sol | |
| D | silica sol | $H_2O$ | MgO, Δ | $Al(H_2PO_4)_3$ | |
| E | silica sol | MgO, Δ | $Al(H_2PO_4)_3$ | $H_2O$ | |
| F | $Al(H_2PO_4)_3$ | MgO, Δ | silica sol | $H_2O$ | |
| G | $Al(H_2PO_4)_3$ | MgO | $H_2O$ | silica sol | |
| H | $Al(H_2PO_4)_3$ | MgO, Δ | $Al(H_2PO_4)_3$ | silica sol | $H_2O$ |

[1] The paint must be stirred to uniformity before proceeding to the next step. Δ The paint needs to be heated in the step noted with Δ.

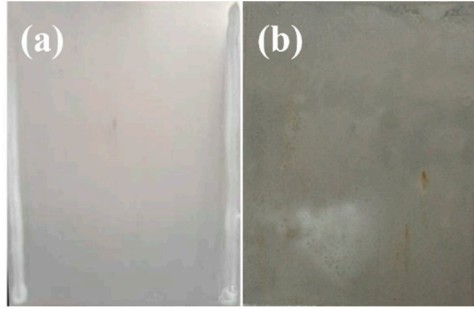

**Figure 12.** The morphology of sample prepared by Method H: (**a**) new sample and (**b**) after NSS for 72 h.

To improve the structure and performance of the coating, not only was the MgO used, but the lithium silicate ($Li_2SiO_3$) and potassium silicate ($K_2SiO_3$) were also used. $Li_2SiO_3$ and $K_2SiO_3$ are generally considered to form Si-O-Si bonds, which are like the characteristic bonds of silica sol. In addition, $Li_2SiO_3$ has self-curing ability, which can improve the curing conditions of the coating. Therefore, we separately designed chromium-free inorganic coatings containing $Li_2SiO_3$ and $K_2SiO_3$.

$Li_2SiO_3$ and $K_2SiO_3$ are water-soluble and can be directly added to chromium-free insulation coating. Therefore, the research is mainly aimed at the influence of the $Li_2SiO_3$ or $K_2SiO_3$ content on the performance. Table 7 demonstrates the detailed composition of coatings with different content of $Li_2SiO_3$ or $K_2SiO_3$.

The two kinds of silica sol with different particle sizes used in 1% Li-Ac are acidic silica sol, while those in other coatings are acidic silica sol and alkaline silica sol. The 5% Li and 5% K have poor stability and gel quickly. The ungelled coatings are used on the oriented silicon-steel sheets for the neutral salt-spray resistance test. The morphology of the new board and the morphology after 72 h of neutral salt-spray resistance are shown in Figure 13.

**Table 7.** The coating with different content of $Li_2SiO_3$ or $K_2SiO_3$.

| Sample | $Al(H_2PO_4)_3$ (g) | Silica Sol (g) | $Li_2SiO_3$ (g) | $K_2SiO_3$ (g) | Deionized Water (g) | MgO (g) |
|---|---|---|---|---|---|---|
| 5% Li | 35 | 40 | 5 | - | 20 | - |
| 1% Li-Ac | 35 | 40 | 1 | - | 24 | - |
| 0.5% K | 35 | 40 | - | 0.5 | 24.5 | - |
| 1% K | 35 | 40 | - | 1 | 24 | - |
| 2% K | 35 | 40 | - | 2 | 23 | - |
| 5% K | 35 | 40 | - | 5 | 20 | - |
| 5% K-Mg | 35 | 40 | - | 5 | 19.5 | 0.5 |

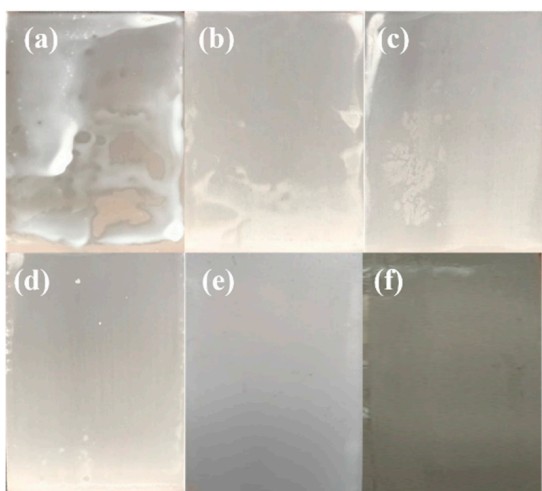

**Figure 13.** The morphology of sample with different coatings: (**a**) 1% Li-Ac, (**b**) 0.5% K, (**c**) 1% K, (**d**) 2% K, (**e**) 5% K-Mg and (**f**) 5% K-Mg after NSS for 72 h.

Many shrinkage holes and shrinkage edges appeared before the 1% Li-Ac coating is cured. After curing, the shrinkage area is increased, and the color of the surface is uneven (Figure 13a). The results indicate that the $Li_2SiO_3$ is unsuitable for the coating since it might change the tension. There are obvious shrinkage holes and shrinkage edges on the 0.5% K (Figure 13b) and 1% K (Figure 13c). The plate surface is not uniform. Although no particles are observed, there are white spots and their trailing traces on the surface of the 2% K sample (Figure 13d), indicating that there are gel particles in the coating. It also proves that the stability of 2% K is worse than that of 0.5% K and 1% K. 5% K-Mg has few precipitations, but there is no obvious defect on the plate prepared (Figure 13e), and the corrosion-resistance area of the NSS for 72 h is less than 5% (Figure 13f), so $K_2SiO_3$ is more suitable for chromium-free insulating coating.

Since $Li_2SiO_3$ is unsuitable for chromium-free insulating coating, in order to adjust the pH value to a stable range and add lithium ions to improve the damp-heat resistance and density of the coating, LiOH was introduced into the chromium-free insulating coating. Referring to the method of adding MgO, LiOH was dissolved in $Al(H_2PO_4)_3$ solution to prepare $Al(H_2PO_4)_3$-LiOH solution with 10 wt.% LiOH, and was used in chromium-free insulating coatings. The formula information is shown in Table 8.

**Table 8.** The insulating coatings containing LiOH.

| Sample | Deionized Water (g) | $Al(H_2PO_4)_3$ (g) | $Al(H_2PO_4)_3$-MgO (g) | $Al(H_2PO_4)_3$-LiOH (g) | Silica Sol (g) | |
|---|---|---|---|---|---|---|
| | | | | | Acidic 8 nm | Alkaline 15 nm |
| 0.5% Li-Mg | 24 | 13.5 | 17.5 | 5 | 20 | 20 |
| 0.5% Li | 24.5 | 30.5 | - | 5 | 20 | 20 |

As shown in Figure 13, the appearance of the 0.5% Li-Mg is better (Figure 14a), while the surface of the 0.5% Li is uneven and the shrinkage edges are obvious, resulting in some area exposed (Figure 14b). The NSS was conducted on the 0.5% Li-Mg, and there was no corrosion after 24 h (Figure 14c).

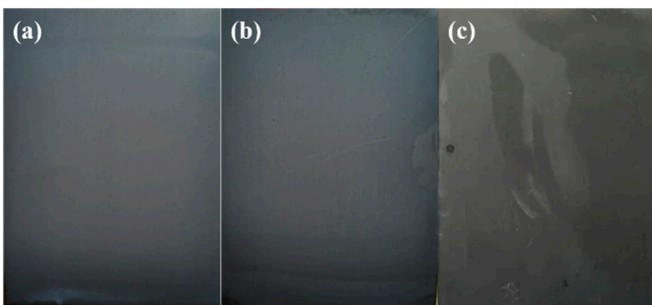

**Figure 14.** The morphology of sample with different coatings: (**a**) 0.5% Li-Mg, (**b**) 0.5% Li and (**c**) 0.5% Li-Mg after NSS for 24 h.

The above results proved that appropriate addition of $K_2SiO_3$ and LiOH can improve the integrity and corrosion resistance of the coating. However, in parallel experiments, the corrosion resistance of the coating including $K_2SiO_3$ or LiOH was not stable, so how to improve the performance of the coating needs further research.

### 4. Conclusions

According to the above research, an environmentally friendly high-temperature resistance coating for oriented silicon steel was designed. The coating is nontoxic, chrome-free and environmentally friendly. The suitable heat-treatment process is 475 °C for 10 s and 800 °C for 40 s, which makes it able to be cured in situ in the rolling process of oriented silicon steel. The coating did not undergo a pyrolysis reaction during the annealing but a phase transition from amorphous to crystalline $AlPO_4$ and $SiO_2$, which helps the coating keep high integrity and coverage and ensures the corrosion resistance of the coating. The main body of the coating is the stacked $SiO_2$ particles, so the corrosion-resistance mechanism of the coating is the shielding effect. Based on the shielding effect. the effect of the coating components on the corrosion resistance of the coating was systematically investigated, including the particle size of silica sol, the addition of metal oxides and silicates, etc., and finally, the corrosion area was less than 5% after 72 h in the NSS test, which is better than the industry standard. In the adjustment process, the improvement of corrosion resistance is achieved by the synergistic effect of each component, such as the combination of different particle sizes of $SiO_2$, through which the penetration path of the corrosive medium will be more complicated and longer. However, the stability of the coating still needs to be improved. Highly stable coatings with excellent corrosion resistance and the structural changes in the heat process will be the focus of further research.

**Author Contributions:** Conceptualization, Y.L.; methodology, Y.L. and A.C.; formal analysis, A.C., Y.Z. and Z.L.; investigation, Y.L. and X.Y.; resources, L.W., H.F. and Y.H.; writing—original draft preparation, X.Y. and B.Z.; writing—review and editing, Y.L., C.X. and Y.H.; supervision, L.W., H.F. and Y.H. All authors have read and agreed to the published version of the manuscript.

**Funding:** This work was financially supported by the Key Research and Development Program of Hubei Province (Grant No. 2020BAB084 and No. 2021BAA063), National Natural Science Foundation of China (Grant No. 61904130), and Key Laboratory of Hubei Province for Coal Conversion and New Carbon Materials (Grant No. WKDM201907).

**Institutional Review Board Statement:** Not applicable.

**Informed Consent Statement:** Not applicable.

**Data Availability Statement:** The data that support the findings of this study are available from the corresponding author upon reasonable request.

**Acknowledgments:** We would like to thank Wang at the Analytical & Testing Center of Wuhan University of Science and Technology for the help on XRD analysis.

**Conflicts of Interest:** The authors declare no conflict of interest. The funders had no role in the design of the study; in the collection, analyses, or interpretation of data; in the writing of the manuscript; or in the decision to publish the results.

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
