# Peer review of "Component Design of Environmentally Friendly High-Temperature Resistance Coating for Oriented Silicon Steel and Effects on Anti-Corrosion Property"

_coatings, doi:10.3390/coatings12070959_

Round 1
Reviewer 1 Report
The organization and presentation of work is good in the present manuscript using the TGA analysis, XRD, SEM and corrosion test. However, the authors need to address the following issues:
1. In XRD analysis, there is not much difference in the XRD patterns of dried at 60oC and annealed at 475oC, Justify.
2. The authors assigned the same peak to both AlPO4 and SiO2, Justify.
3. Compare the XRD data with JCPDS card data.
4. Image quality is very poor, provide the quality images.
5. The conclusion section needs to be improved in a scientific manner.
6. How can you improve the coating characteristics?
Author Response
Response to Reviewer 1 Comments
Point 1: In XRD analysis, there is not much difference in the XRD patterns of dried at 60oC and annealed at 475 oC, Justify.
Response 1: Thank you for your suggestion. We make several parallel tests and all the experimental results show that the XRD patterns of dried at 60oC and annealed at 475 oC are highly similar. We believe it’s formed by multiple factors. First of all, the heating time at 475°C is short and the products of phase transformation are less, which means the peak intensity would be weak. Secondly, comparing to the temperature of crystal transition for aluminum phosphate (AlPO4) and silicon dioxide (SiO2), the temperature of 475°C is slightly lower. The main composition of our coating is aluminum dihydrogen phosphate(Al(H2PO4)3) and silica sol, their dehydration products are AlPO4 and SiO2. According to TGA, it can be found that the quality has basically reached stability after 300 oC, and there is no significant weight loss after that, which indicates that most free water and bound water have been removed and the main components of coating are AlPO4 and SiO2. According to most reports, the temperature-promoted phase transition of AlPO4 mostly occurs at about 450-500°C, and the temperature for SiO2 is about 500-600°C ([1] Jia, J.Y.; Luo, L.L.; Bai, J.W. Effect of aluminum dihydrogen phosphate on properties of Al2O3-SiO2 system castables. Ceramics 2008, 29-31. https://doi.org/10.19397/j.cnki.ceramics.2008.05.007. [2] Liu, J.B.; Yuan, Z.Z.; Lu, S.S.; Zhou, Z.G.; Jia, S.W. Study on Al(H2PO4)3 strengthening bauxite clinker shell. Foundry 2016, 65, 258-261,266. https://doi.org/10.3969/j.issn.1001-4977.2016.03.011. [3] Wang, X.B.; Li, F.; Deng, C.H.; Wang, B.C. Preparation of white fused alumina-silica sol coating on 310s stainless steel and study on its performance. Electroplating and Finishing 2017, 36, 328-331. https://doi.org/10.19289/j.1044-227x.2017.06.013. [4] Yao, S.; Lin, H.Y.; Zeng, X.Z.; Zhuang, Y.H.; Li, C.Y.; Li, Y. Preparation of porous alumina ceramics by silica solution bonding method. Journal of Ceramics 2019, 40, 325-329. https://doi.org/10.13957/j.cnki.tcxb.2019.03.009.). This is also the reason why the peak shape on the XRD pattern at 475 °C is not significant. Finally, during the sample preparation, the moisture in the coating has been removed by drying at 60°C, so the peak position of XRD pattern at 60°C is consistent with the subsequent pattern, but it’s broader.
Point 2: The authors assigned the same peak to both AlPO4 and SiO2, Justify.
Response 2: Thank you for your suggestion. We assigned the same peaks in the XRD pattern to both AlPO4 and SiO2 based on the combination of experimental analysis results and literature survey results. Taking AlPO4 (ICDD/JCPDS 11-0500) and SiO2 (ICDD/JCPDS 27-0605) as examples, the strongest characteristic peaks for AlPO4 and SiO2 are 21.781° and 21.604°, respectively. They are close. In our actual results, there are characteristic peaks around 21.6°-21.7° in the patterns at 475°C and 800°C, and there are also characteristic peaks near 35.6°-35.7° in the pattern at 800°C. Due to instrument limitations, we cannot make a more precise distinction between the two species. What’s more, according to the reports, AlPO4 and SiO2 have a wide temperature range for the formation of crystal phases, though the temperature for SiO2 is higher. We cannot affirm that there is no crystal phase of SiO2 at 475 °C by sample preparation conditions. In fact, there are also reports about that the peak positions of AlPO4 and SiO2 are overlaped in the XRD patterns ([5] Wang, C.; Wang, K.; Ding, H.; Xiao, X.; Wang, B.; Chen, J.; Zhang, W.; Zhao, L.; Yu, X. Effects of ammonium tungstate on the properties of insulating coating for grain-oriented silicon steel. Materials Science and Engineering: B 2018, 228, 109–116. https://doi.org/10.1016/j.mseb.2017.11.022. [6] Luo, X.D.; Zhan, H.S.; Li, Y.J.; Xie, Z.P.; Wu, Y.X. Effects of cement types on properties of high-alumina wear-resistant plastic refractories. Naihuo Cailiao 2015, 49, 255-258, 263. https://doi.org/10.3969/j.issn.1001-1935.2015.05.004.). Therefore, on the basis that the existence of SiO2 crystal phase in the coating after treatment at 475 °C is possible, we drew the XRD pattern as shown in Figure 2. Combined with the composition of the coating, we believe that there are AlPO4 (111) and SiO2 (111) in the coating after treatment at 475 °C and AlPO4 (111) would be more than the other. This is because the characteristic peak position is more inclined to AlPO4 (111). And there are more SiO2 (111) in the coating after treatment at 800 °C for the characteristic peak position is more inclined to SiO2 (111) and the content of silica sol in the coating is higher than that of Al(H2PO4)3.
Point 3: Compare the XRD data with JCPDS card data.
Response 3: Thank you for your suggestion. The number of card data has been improved in Figure 2, and a description has also been added to the text and highlighted in yellow.
Point 4: Image quality is very poor, provide the quality images.
Response 4:Thank you for your suggestion. We have updated the images in the manuscript, adjusted settings to avoid loss of clarity due to compression, and uploaded image source files in the submission system.
Point 5: The conclusion section needs to be improved in a scientific manner.
Response 5: Thank you for your suggestion. The conclusions section has been revised to focus more on the conclusions and the analysis of the mechanisms involved, rather than the data and results obtained.
Point 6: How can you improve the coating characteristics?
Response 6: Thank you for your suggestion. In this manuscript, the improvement of coating performance is achieved through two processes:
1. Analyze the structural characteristics and mechanism of the coating.
According to the comprehensive analysis of SEM, TGA and XRD, it is found that the coating achieves corrosion resistance through shielding. The test results of TGA show that the coating has little weight loss at high temperature, and still retains most of the mass after heat treatment. The test results of XRD verify that the coating does not undergo weight loss reactions such as decomposition at high temperature, but phase transition, which ensures the integrity of the coating. By SEM, it can be observed that the coating surface is obviously granular, and there is no exposed substrate material. The larger Nyquist radius and lower corrosion current density in electrochemical tests also indicate that the coating has a good shielding effect and can slow down the corrosion rate.
2. Adjust the coating formula according to the mechanism of the coating.
Based on the analysis of how the coating behaves, we made adjustments to the coating formulation. Firstly, according to the characteristics that the coating is formed by the accumulation of spherical particles, we adjusted the particle size of the silica sol, and tried to make the corrosion penetration path more complicated through the mechanism shown in Figure 8 in our manuscript. In this process, we optimized the storage stability of the coating by adjusting the pH of the silica sol, and evaluated the impact of the pH on the corrosion resistance of the coating. Secondly, we try to optimize the properties of the coating by adding additives, including foam and leveling equality, etc., which help the coating to form a more uniform film and optimize the shielding effect. Finally, we used MgO, Li2SiO3, K2SiO3 and LiOH as functional fillers for the coating, respectively. Among them, MgO is considered to accelerate the condensation rate of Al(H2PO4)3 and form a denser network structure in some reports ([7] Zhang, H.J. Preparation and corrosion resistance of phosphate coatings on Q235 steel surface. Master degree thesis, Harbin Institute of Technology, 2020, https://doi.org/10.27061/d.cnki.ghgdu.2020.003314. ). Li2SiO3 and K2SiO3 can improve the silica system to form a glass phase structure. LiOH is regarded as a kind of the Li+ source for reacting with the silica sol to form Li2SiO3. Not all of the above attempts have achieved positive results. After the combination of the best, a chromium-free inorganic coating for oriented silicon steel that is resistant to high temperature and neutral salt spray test for 72 hours is obtained. However, the study also shows that through optimization, the properties of the coating can be further improved.

Reviewer 2 Report
The authors performed the studies on “Component Design of Environmentally Friendly High-temperature Resistance Coating for Oriented Silicon Steel and Effects on Anti-corrosion Property”. The results obtained in this study is interesting. However, the authors may be address the following comments and incorporate the same into the manuscript for the reader point of view for better understanding.
- Kindly mention aluminum dihydrogen phosphate (Al(H2PO4)3) in line no 73 in full form, because in the line no 73 only you have used the word first time instead of line no 77.
- What is the novelty of the present research work, Since Al(H2PO4)3 and silica sol are reported in the past also.
- Kindly write the full form at first instant like NSS, TGA, DTG etc. It may be familiar terms, even though it may not aware for the researchers who start work newly to the field.
- Section 2.2 Coating Preparation. According to the formula….. what is the formula authors adopted in this study for mixing the sample. Is there any ration is followed to mix water with silica sol.
- Section 2,2, how authors confirmed the coatings are uniform
- Why the authors performed the study at Nitrogen atmosphere, why not other gas like argon or without any gas like air.
- How the authors are ensuring the coatings are porous free,
- Why the authors are not performed the SEM/EDS analysis in the present study after the coating was carried out
- Why the authors are not performed the XRD analysis to evaluate the formation of different phases on the substrate
- Authors are not measured the mass loss/gain after the samples are exposed to the environment
- In the present study the authors performed the study in 3.5%NaCL environment. Is the samples are exposed in the 3.5%NaCL environment in the real time application
- Why the corrosion mechanism are not derived in the present investigation
- Section title 3, should be rewrite as Results and Discussion
Author Response
Response to Reviewer 2 Comments
Point 1: Kindly mention aluminum dihydrogen phosphate (Al(H2PO4)3) in line no 73 in full form, because in the line no 73 only you have used the word first time instead of line no 77.
Response 1: Thank you for your suggestion. We have revised the statements of aluminum dihydrogen phosphate(Al(H2PO4)3) in lines 73 and 77 in the manuscript, and highlighted it in yellow in the revised manuscript.
Point 2: What is the novelty of the present research work, Since Al(H2PO4)3 and silica sol are reported in the past also.
Response 2: Thank you for your suggestion. In the previous reports, especially the reports of high temperature adhesives, the respective use of Al(H2PO4)3 and silica sol is more common. In the field of anti-corrosion coating, there are also some reports about the research of Al(H2PO4)3 or silica sol as an auxiliary film-forming agent. However, the organic resin as the main film-forming material is still the mainstream of the research on the chromium-free coating for the oriented silicon steel. In the few reports about chromium-free inorganic coating for the oriented silicon steel, Al(H2PO4)3 is the main film-forming material. And the poor corrosion resistance of such inorganic coatings has become a widespread understanding of the performance evaluation for chromium-free inorganic coating. In our research, Al(H2PO4)3 and silica sol are combined and used as the main film-forming material. Based on the shielding effect of the coating, silica sol becomes SiO2 as the main body of the coating, Al(H2PO4)3 becomes AlPO4 as the auxiliary. And combined the optimization of the formula, we finally achieve the both high temperature resistance and corrosion resistance of the coating.
Point 3: Kindly write the full form at first instant like NSS, TGA, DTG etc. It may be familiar terms, even though it may not aware for the researchers who start work newly to the field.
Response 3: Thank you for your suggestion. We have added the full form of TGA and NSS in Section 2.3, added the full form of TG, DTG and DSC in line no 140 in Section 3 and highlighted in yellow in the manuscript.
Point 4: Section 2.2 Coating Preparation. According to the formula….. what is the formula authors adopted in this study for mixing the sample. Is there any ration is followed to mix water with silica sol.
Response 4: Thank you for your suggestion.The core research content of this manuscript is to report a high-temperature-resistant annd corrosion-resistant chromium-free inorganic coating for oriented silicon steel and the component design process of the coating based on the corrosion resistance. In each stage of component design process, the selection of raw materials and the specific composition of the coating are different. However. in consideration of environmental protection and ease of operation, the selection of raw materials is basically based on the principle of harmlessness and solubility. Therefore, although the composition of the coating is different, the preparation operation of the coating is basically the same. In Section 2.2, the description of coating preparation, based on this point, we only stated the preparation operation but did not specify the specific composition of the coating. However, in Section 3, the specific discussion on various factors, we have lists to indicate the specific composition of the coating, which also includes the ratio of water and silica sol.
Point 5: Section 2,2, how authors confirmed the coatings are uniform.
Response 5: Thank you for your suggestion. In the coating stage, we use tools to ensure uniform film. We use a No. 6 wire-wound rod coater on which the metal wires with the same diameter are wound or die-casting on the metal rod. When the operator simulates the roller shaft for coating, the coating retained in the gap between the metal wires would be leveling to form the film. When the wettability and leveling performance of the coating are qualified and the operator's force is appropriate, the film will be uniform, and there will be no obvious defect.
Point 6: Why the authors performed the study at Nitrogen atmosphere, why not other gas like argon or without any gas like air
Response 6: Thank you for your suggestion. In the TGA analysis, Nitrogen atmosphere is used because the curing and annealing of the insulating coating are carried out with the protection of Nitrogen in the manufacture of oriented silicon steel. With such condition, the loss of the coating containing organic components is more due to thermal decomposition than oxidation or combustion reaction. Therefore, to simulate the actual manufacture environment, the TGA analysis of semi-inorganic coating and inorganic coating was carried out in Nitrogen atmosphere. And the significant weight loss phenomenon of semi-inorganic coating can still be observed.
Point 7: How the authors are ensuring the coatings are porous free,
Response 7: Thank you for your suggestion. The holes mentioned in the manuscript have two meanings. The first one is the macroscopic visible holes during the coating preparation stage, which refer to directly expose the substrate, caused by errors in coating operations or poor coating leveling. Therefore, it is necessary to adjust the formula and operations so that the wet film can be porous free. The second is the micro pores observed by the SEM analysis. The pore diameter is small and about 50-200nm. Since the coating is formed by the accumulation of SiO2 particles, it is difficult to avoid this kind of micro pore. Some small holes can also be observed in the SEM images we present. This kind of small hole does not mean the direct exposure of the substrate, but the increase of the penetration inlet for corrosive medium. Therefore, we try to reduce this kind of micro pore by adjusting the components such as the particle size of silica sol and addition of MgO.
Point 8: Why the authors are not performed the SEM/EDS analysis in the present study after the coating was carried out
Response 8: Thank you for your suggestion. Although the corrosion resistance is related to the coating morphology, the neutral salt spray test (NSS) and electrochemical test are more direct evaluation methods for the corrosion resistance. Therefore, we mainly use NSS and electrochemical test to test the coating performance. However, to analyze the mechanism of the coating, we used SEM to observe the best coating we prepared, and took the blank plate as comparison. This comparison is to show that the coating completely covers the substrate. At the same time, we also found that the coating is formed by the accumulation of particles through SEM, which supports the analysis of the mechanism of the coating. In addition, the blank plate contains magnesium silicate (MgSiO3) mezzanine characterized by uneven morphology and exposure of the metal substrate, while the surface of coated plate is flat without prominent pores. There is a significant contrast from the morphology for judgment. Although we have carried out EDS analysis, the results show that the elements in the blank plate are Si, O, Mg and Fe, while the coated plate is Si, O, Al, P and Mg, which can only prove the coverage of the coating and does not further promote the analysis of the mechanism. Therefore, we only present Figure 4 in the manuscript as the SEM results, without adding EDS results.
Point 9: Why the authors are not performed the XRD analysis to evaluate the formation of different phases on the substrate
Response 9: Thank you for your suggestion. In the process of designing the experimental methods, our initial phase observation strategy was to coat the same coating on the same batch of oriented silicon steel sheets, and conduct XRD analysis after different heat treatments. The results show that it is difficult to exclude the signal of the oriented silicon steel substrate even with the glancing incidence, since the the coating thickness is only 1-2 μm while the substrate is 270μm. The signal of the substrate is too strong to analyze the signal of the coating. We also try to scrape the coating from the coated plate. But it’s impossible to ensure that the MgSiO3 mezzanine will not be scraped off, since the hardness of the coating is high and the thickness is small. Finally, we choose to prepare the coating powder for XRD analysis.
Point 10: Authors are not measured the mass loss/gain after the samples are exposed to the environment
Response 10: Thank you for your suggestion. We did not present weight change monitoring in the manuscript for two reasons:
- We have used electrochemical test and NSS to evaluate the corrosion resistance. Electrochemical test can sensitively observe the micro electrochemical reaction at the interface between the coating and the substrate, and can effectively and rapidly evaluate the corrosion resistance of the coating. NSS is a standardized operation based on national and international standards, which provides a horizontal comparison possibility with high reliability in research and industrial application. In the NSS, the corrosion area counted is the final evaluation method, so we have recorded the corrosion area rather than the quality change.
- The oriented silicon steel sheet is a sandwich structure. In the process of corrosion deepening and diffusion, there is not only the peeling of the coating but also the peeling of the MgSiO3 mezzanine. The description of this phenomenon will be more detailed in our other manuscripts. However, due to this phenomenon, the accuracy of the weight change monitoring for the oriented silicon steel during the corrosion process is less than that of other metal materials such as stainless steel and Q235 steel.
In the future research, we will look for more appropriate experimental strategies to achieve effective and accurate weight change monitoring.
Point 11: In the present study the authors performed the study in 3.5%NaCL environment. Is the samples are exposed in the 3.5%NaCL environment in the real time application
Response 11: Thank you for your suggestion. The use of 3.5% NaCl solution in electrochemical analysis is based on reports about anti-corrosion coating and our experimental needs.The actual service environment of the oriented silicon steel is as core laminations in transformers, motors or generators. Most of the corrosion occures in the storage and processing of steel and is the atmospheric corrosion. The 3.5% NaCl solution is similar to the marine environment. It is also the Cl- concentration region where the metal corrosion rate is faster according to reports([1] Wang, X.; Xiao, K.; Cheng, X.Q.; Dong, C.F.; Wu, J.S.; Yi, P.; Mao, C.L.; Jiang, L.; Li, X.G. Corrosion prediction mode of Q235 steel in polluted marine atmospheric enviroment. Journal of Materials Engineering 2017, 45, 51-57. https://doi.org/10.11868/j.issn.1001-4381.2015.001414. [2] Wang, H.T.; Han, E.H.; Ke, W. Gray model and gray relation analysis for atmospheric corrosion of carbon steel and low steel. Corrosion Science and Protection Technology 2006,18,278-280. https://doi.org/10.3969/j.issn.1002-6495.2006.04.012. ). And in the reports about anti-corrosion coating, the 3.5% NaCl solution is widely used as the electrolyte solution for electrochemical analysis to test the the anti-corrosion performance of organic and semi-inorganic coatings ([3] Wang, S.H.; Long, Y.F.; Zhao, S.L.; An, C.Q.; Ding, K. A New Chromate-free insulating coating on Silicon Steel. Advanced Materials Research 2014, 1004–1005, 757–762. https://doi.org/10.4028/www.scientific.net/AMR.1004-1005.757. [4] Shi, C.; Shao, Y.; Wang, Y.; Meng, G.; Liu, B. Influence of submicro-sheet zinc phosphate modified by urea-formaldehyde on the corrosion protection of epoxy coating. Surfaces and Interfaces 2020, 18, 100403. https://doi.org/10.1016/j.surfin.2019.100403. [5] Wang, Y. N.; Dai, X. Y.; Xu, T. L.; Qu, L. J.; Zhang, C. L. Preparation and anticorrosion properties of saline grafted nano-silica/epoxy composite coating. Chemical Journal of Chinese Universities 2018, 39, 1564-1572. https://doi.org/10.7503/cjcu20170821.). The use of the same solution facilitates the performance comparison of our coatings with those reported. In addition, this concentration is lower than the NaCl concentration in NSS. This is because we consider that the solution in NSS is in the spraying state and does not stay on the plate surface, while the plate surface acting as the working electrode in the electrochemical test and needs to be immersed in the solution. In order to meet the needs of our other experimental strategies, we finally chose 3.5% NaCl as the electrochemical test environment.
Point 12: Why the corrosion mechanism are not derived in the present investigation
Response 12: Thank you for your suggestion. In this manuscript, we mainly present the anti-corrosion mechanism of the coating, namely the shielding effect. And we focus more on the influence of coating components adjustment, such as the change of silica sol particle size, on the shielding effect of the coating, that is, the action mechanism shown in Figure 8 in the manuscript. Through TGA, XRD and SEM analysis, it can be speculated that the structure of the coating is the combination of AlPO4 which provides structural stability through a three-dimensional network structure and stacked SiO2 particles. And it still has a complete structure to realize effec-tive coverage after high-temperature annealing. For the coating based on the shielding effect, making the path of the corrosive more tortuous will be an effective way to improve corrosion resistance. Based on this analysis, we take the SiO2 as the example and summarize the effect of coating components on coating performance as shown in Figure 8 in the manuscript.
Figure 8. Scheme of the effect of silica sol particle size on the corrosion resistance
Response to Reviewer 2 Comments
Point 1: Kindly mention aluminum dihydrogen phosphate (Al(H2PO4)3) in line no 73 in full form, because in the line no 73 only you have used the word first time instead of line no 77.
Response 1: Thank you for your suggestion. We have revised the statements of aluminum dihydrogen phosphate(Al(H2PO4)3) in lines 73 and 77 in the manuscript, and highlighted it in yellow in the revised manuscript.
Point 2: What is the novelty of the present research work, Since Al(H2PO4)3 and silica sol are reported in the past also.
Response 2: Thank you for your suggestion. In the previous reports, especially the reports of high temperature adhesives, the respective use of Al(H2PO4)3 and silica sol is more common. In the field of anti-corrosion coating, there are also some reports about the research of Al(H2PO4)3 or silica sol as an auxiliary film-forming agent. However, the organic resin as the main film-forming material is still the mainstream of the research on the chromium-free coating for the oriented silicon steel. In the few reports about chromium-free inorganic coating for the oriented silicon steel, Al(H2PO4)3 is the main film-forming material. And the poor corrosion resistance of such inorganic coatings has become a widespread understanding of the performance evaluation for chromium-free inorganic coating. In our research, Al(H2PO4)3 and silica sol are combined and used as the main film-forming material. Based on the shielding effect of the coating, silica sol becomes SiO2 as the main body of the coating, Al(H2PO4)3 becomes AlPO4 as the auxiliary. And combined the optimization of the formula, we finally achieve the both high temperature resistance and corrosion resistance of the coating.
Point 3: Kindly write the full form at first instant like NSS, TGA, DTG etc. It may be familiar terms, even though it may not aware for the researchers who start work newly to the field.
Response 3: Thank you for your suggestion. We have added the full form of TGA and NSS in Section 2.3, added the full form of TG, DTG and DSC in line no 140 in Section 3 and highlighted in yellow in the manuscript.
Point 4: Section 2.2 Coating Preparation. According to the formula….. what is the formula authors adopted in this study for mixing the sample. Is there any ration is followed to mix water with silica sol.
Response 4: Thank you for your suggestion.The core research content of this manuscript is to report a high-temperature-resistant annd corrosion-resistant chromium-free inorganic coating for oriented silicon steel and the component design process of the coating based on the corrosion resistance. In each stage of component design process, the selection of raw materials and the specific composition of the coating are different. However. in consideration of environmental protection and ease of operation, the selection of raw materials is basically based on the principle of harmlessness and solubility. Therefore, although the composition of the coating is different, the preparation operation of the coating is basically the same. In Section 2.2, the description of coating preparation, based on this point, we only stated the preparation operation but did not specify the specific composition of the coating. However, in Section 3, the specific discussion on various factors, we have lists to indicate the specific composition of the coating, which also includes the ratio of water and silica sol.
Point 5: Section 2,2, how authors confirmed the coatings are uniform.
Response 5: Thank you for your suggestion. In the coating stage, we use tools to ensure uniform film. We use a No. 6 wire-wound rod coater on which the metal wires with the same diameter are wound or die-casting on the metal rod. When the operator simulates the roller shaft for coating, the coating retained in the gap between the metal wires would be leveling to form the film. When the wettability and leveling performance of the coating are qualified and the operator's force is appropriate, the film will be uniform, and there will be no obvious defect.
Point 6: Why the authors performed the study at Nitrogen atmosphere, why not other gas like argon or without any gas like air
Response 6: Thank you for your suggestion. In the TGA analysis, Nitrogen atmosphere is used because the curing and annealing of the insulating coating are carried out with the protection of Nitrogen in the manufacture of oriented silicon steel. With such condition, the loss of the coating containing organic components is more due to thermal decomposition than oxidation or combustion reaction. Therefore, to simulate the actual manufacture environment, the TGA analysis of semi-inorganic coating and inorganic coating was carried out in Nitrogen atmosphere. And the significant weight loss phenomenon of semi-inorganic coating can still be observed.
Point 7: How the authors are ensuring the coatings are porous free,
Response 7: Thank you for your suggestion. The holes mentioned in the manuscript have two meanings. The first one is the macroscopic visible holes during the coating preparation stage, which refer to directly expose the substrate, caused by errors in coating operations or poor coating leveling. Therefore, it is necessary to adjust the formula and operations so that the wet film can be porous free. The second is the micro pores observed by the SEM analysis. The pore diameter is small and about 50-200nm. Since the coating is formed by the accumulation of SiO2 particles, it is difficult to avoid this kind of micro pore. Some small holes can also be observed in the SEM images we present. This kind of small hole does not mean the direct exposure of the substrate, but the increase of the penetration inlet for corrosive medium. Therefore, we try to reduce this kind of micro pore by adjusting the components such as the particle size of silica sol and addition of MgO.
Point 8: Why the authors are not performed the SEM/EDS analysis in the present study after the coating was carried out
Response 8: Thank you for your suggestion. Although the corrosion resistance is related to the coating morphology, the neutral salt spray test (NSS) and electrochemical test are more direct evaluation methods for the corrosion resistance. Therefore, we mainly use NSS and electrochemical test to test the coating performance. However, to analyze the mechanism of the coating, we used SEM to observe the best coating we prepared, and took the blank plate as comparison. This comparison is to show that the coating completely covers the substrate. At the same time, we also found that the coating is formed by the accumulation of particles through SEM, which supports the analysis of the mechanism of the coating. In addition, the blank plate contains magnesium silicate (MgSiO3) mezzanine characterized by uneven morphology and exposure of the metal substrate, while the surface of coated plate is flat without prominent pores. There is a significant contrast from the morphology for judgment. Although we have carried out EDS analysis, the results show that the elements in the blank plate are Si, O, Mg and Fe, while the coated plate is Si, O, Al, P and Mg, which can only prove the coverage of the coating and does not further promote the analysis of the mechanism. Therefore, we only present Figure 4 in the manuscript as the SEM results, without adding EDS results.
Point 9: Why the authors are not performed the XRD analysis to evaluate the formation of different phases on the substrate
Response 9: Thank you for your suggestion. In the process of designing the experimental methods, our initial phase observation strategy was to coat the same coating on the same batch of oriented silicon steel sheets, and conduct XRD analysis after different heat treatments. The results show that it is difficult to exclude the signal of the oriented silicon steel substrate even with the glancing incidence, since the the coating thickness is only 1-2 μm while the substrate is 270μm. The signal of the substrate is too strong to analyze the signal of the coating. We also try to scrape the coating from the coated plate. But it’s impossible to ensure that the MgSiO3 mezzanine will not be scraped off, since the hardness of the coating is high and the thickness is small. Finally, we choose to prepare the coating powder for XRD analysis.
Point 10: Authors are not measured the mass loss/gain after the samples are exposed to the environment
Response 10: Thank you for your suggestion. We did not present weight change monitoring in the manuscript for two reasons:
- We have used electrochemical test and NSS to evaluate the corrosion resistance. Electrochemical test can sensitively observe the micro electrochemical reaction at the interface between the coating and the substrate, and can effectively and rapidly evaluate the corrosion resistance of the coating. NSS is a standardized operation based on national and international standards, which provides a horizontal comparison possibility with high reliability in research and industrial application. In the NSS, the corrosion area counted is the final evaluation method, so we have recorded the corrosion area rather than the quality change.
- The oriented silicon steel sheet is a sandwich structure. In the process of corrosion deepening and diffusion, there is not only the peeling of the coating but also the peeling of the MgSiO3 mezzanine. The description of this phenomenon will be more detailed in our other manuscripts. However, due to this phenomenon, the accuracy of the weight change monitoring for the oriented silicon steel during the corrosion process is less than that of other metal materials such as stainless steel and Q235 steel.
In the future research, we will look for more appropriate experimental strategies to achieve effective and accurate weight change monitoring.
Point 11: In the present study the authors performed the study in 3.5%NaCL environment. Is the samples are exposed in the 3.5%NaCL environment in the real time application
Response 11: Thank you for your suggestion. The use of 3.5% NaCl solution in electrochemical analysis is based on reports about anti-corrosion coating and our experimental needs.The actual service environment of the oriented silicon steel is as core laminations in transformers, motors or generators. Most of the corrosion occures in the storage and processing of steel and is the atmospheric corrosion. The 3.5% NaCl solution is similar to the marine environment. It is also the Cl- concentration region where the metal corrosion rate is faster according to reports([1] Wang, X.; Xiao, K.; Cheng, X.Q.; Dong, C.F.; Wu, J.S.; Yi, P.; Mao, C.L.; Jiang, L.; Li, X.G. Corrosion prediction mode of Q235 steel in polluted marine atmospheric enviroment. Journal of Materials Engineering 2017, 45, 51-57. https://doi.org/10.11868/j.issn.1001-4381.2015.001414. [2] Wang, H.T.; Han, E.H.; Ke, W. Gray model and gray relation analysis for atmospheric corrosion of carbon steel and low steel. Corrosion Science and Protection Technology 2006,18,278-280. https://doi.org/10.3969/j.issn.1002-6495.2006.04.012. ). And in the reports about anti-corrosion coating, the 3.5% NaCl solution is widely used as the electrolyte solution for electrochemical analysis to test the the anti-corrosion performance of organic and semi-inorganic coatings ([3] Wang, S.H.; Long, Y.F.; Zhao, S.L.; An, C.Q.; Ding, K. A New Chromate-free insulating coating on Silicon Steel. Advanced Materials Research 2014, 1004–1005, 757–762. https://doi.org/10.4028/www.scientific.net/AMR.1004-1005.757. [4] Shi, C.; Shao, Y.; Wang, Y.; Meng, G.; Liu, B. Influence of submicro-sheet zinc phosphate modified by urea-formaldehyde on the corrosion protection of epoxy coating. Surfaces and Interfaces 2020, 18, 100403. https://doi.org/10.1016/j.surfin.2019.100403. [5] Wang, Y. N.; Dai, X. Y.; Xu, T. L.; Qu, L. J.; Zhang, C. L. Preparation and anticorrosion properties of saline grafted nano-silica/epoxy composite coating. Chemical Journal of Chinese Universities 2018, 39, 1564-1572. https://doi.org/10.7503/cjcu20170821.). The use of the same solution facilitates the performance comparison of our coatings with those reported. In addition, this concentration is lower than the NaCl concentration in NSS. This is because we consider that the solution in NSS is in the spraying state and does not stay on the plate surface, while the plate surface acting as the working electrode in the electrochemical test and needs to be immersed in the solution. In order to meet the needs of our other experimental strategies, we finally chose 3.5% NaCl as the electrochemical test environment.
Point 12: Why the corrosion mechanism are not derived in the present investigation
Response 12: Thank you for your suggestion. In this manuscript, we mainly present the anti-corrosion mechanism of the coating, namely the shielding effect. And we focus more on the influence of coating components adjustment, such as the change of silica sol particle size, on the shielding effect of the coating, that is, the action mechanism shown in Figure 8 in the manuscript. Through TGA, XRD and SEM analysis, it can be speculated that the structure of the coating is the combination of AlPO4 which provides structural stability through a three-dimensional network structure and stacked SiO2 particles. And it still has a complete structure to realize effec-tive coverage after high-temperature annealing. For the coating based on the shielding effect, making the path of the corrosive more tortuous will be an effective way to improve corrosion resistance. Based on this analysis, we take the SiO2 as the example and summarize the effect of coating components on coating performance as shown in Figure 8 in the manuscript.
Figure 8. Scheme of the effect of silica sol particle size on the corrosion resistance
Response to Reviewer 2 Comments
Point 1: Kindly mention aluminum dihydrogen phosphate (Al(H2PO4)3) in line no 73 in full form, because in the line no 73 only you have used the word first time instead of line no 77.
Response 1: Thank you for your suggestion. We have revised the statements of aluminum dihydrogen phosphate(Al(H2PO4)3) in lines 73 and 77 in the manuscript and highlighted them in yellow in the revised manuscript.
Point 2: What is the novelty of the present research work, Since Al(H2PO4)3 and silica sol are reported in the past also.
Response 2: Thank you for your suggestion. In the previous reports, especially the reports of high-temperature adhesives, the respective use of Al(H2PO4)3 and silica sol is more common. In the field of anti-corrosion coating, there are also some reports about the research of Al(H2PO4)3 or silica sol as an auxiliary film-forming agent. However, organic resin as the main film-forming material is still the mainstream of the research on the chromium-free coating for the oriented silicon steel. In the few reports about chromium-free inorganic coating for the oriented silicon steel, Al(H2PO4)3 is the main film-forming material. And the poor corrosion resistance of such inorganic coatings has become a widespread understanding of the performance evaluation for chromium-free inorganic coating. In our research, Al(H2PO4)3 and silica sol are combined and used as the main film-forming material. Based on the shielding effect of the coating, silica sol becomes SiO2 as the main body of the coating, and Al(H2PO4)3 becomes AlPO4 as the auxiliary. And combined with the optimization of the formula, we finally achieve both high-temperature resistance and corrosion resistance of the coating.
Point 3: Kindly write the full form at first instant like NSS, TGA, DTG etc. It may be familiar terms, even though it may not aware for the researchers who start work newly to the field.
Response 3: Thank you for your suggestion. We have added the full form of TGA and NSS in Section 2.3, added the full form of TG, DTG and DSC in line no 140 in Section 3 and highlighted it in yellow in the manuscript.
Point 4: Section 2.2 Coating Preparation. According to the formula….. what is the formula authors adopted in this study for mixing the sample. Is there any ration is followed to mix water with silica sol.
Response 4: Thank you for your suggestion. The core research content of this manuscript is to report a high-temperature-resistant and corrosion-resistant chromium-free inorganic coating for oriented silicon steel and the component design process of the coating based on the corrosion resistance. In each stage of the component design process, the selection of raw materials and the specific composition of the coating are different. However. in consideration of environmental protection and ease of operation, the selection of raw materials is basically based on the principle of harmlessness and solubility. Therefore, although the composition of the coating is different, the preparation operation of the coating is basically the same. In Section 2.2, the description of coating preparation, based on this point, we only stated the preparation operation but did not specify the specific composition of the coating. However, in Section 3, the specific discussion on various factors, we have lists to indicate the specific composition of the coating, which also includes the ratio of water and silica sol.
Point 5: Section 2,2, how authors confirmed the coatings are uniform.
Response 5: Thank you for your suggestion. In the coating stage, we use tools to ensure uniform film. We use a No. 6 wire-wound rod coater on which the metal wires with the same diameter are wound or die-casting on the metal rod. When the operator simulates the roller shaft for coating, the coating retained in the gap between the metal wires would be leveling to form the film. When the wettability and leveling performance of the coating are qualified and the operator's force is appropriate, the film will be uniform, and there will be no obvious defect.
Point 6: Why the authors performed the study at Nitrogen atmosphere, why not other gas like argon or without any gas like air
Response 6: Thank you for your suggestion. In the TGA analysis, Nitrogen atmosphere is used because the curing and annealing of the insulating coating are carried out with the protection of Nitrogen in the manufacture of oriented silicon steel. With such condition, the loss of the coating containing organic components is more due to thermal decomposition than oxidation or combustion reaction. Therefore, to simulate the actual manufacture environment, the TGA analysis of semi-inorganic coating and inorganic coating was carried out in Nitrogen atmosphere. And the significant weight loss phenomenon of semi-inorganic coating can still be observed.
Point 7: How the authors are ensuring the coatings are porous free,
Response 7: Thank you for your suggestion. The holes mentioned in the manuscript have two meanings. The first one is the macroscopic visible holes during the coating preparation stage, which refer to directly exposing the substrate, caused by errors in coating operations or poor coating leveling. Therefore, it is necessary to adjust the formula and operations so that the wet film can be porous free. The second is the micro pores observed by the SEM analysis. The pore diameter is small and about 50-200nm. Since the coating is formed by the accumulation of SiO2 particles, it is difficult to avoid this kind of micropore. Some small holes can also be observed in the SEM images we present. This kind of small hole does not mean the direct exposure of the substrate, but the increase of the penetration inlet for corrosive medium. Therefore, we try to reduce this kind of micropore by adjusting the components such as the particle size of silica sol and the addition of MgO.
Point 8: Why the authors are not performed the SEM/EDS analysis in the present study after the coating was carried out
Response 8: Thank you for your suggestion. Although the corrosion resistance is related to the coating morphology, the neutral salt spray test (NSS) and electrochemical test are more direct evaluation methods for the corrosion resistance. Therefore, we mainly use NSS and electrochemical tests to test the coating performance. However, to analyze the mechanism of the coating, we used SEM to observe the best coating we prepared, and took the blank plate as comparison. This comparison is to show that the coating completely covers the substrate. At the same time, we also found that the coating is formed by the accumulation of particles through SEM, which supports the analysis of the mechanism of the coating. In addition, the blank plate contains magnesium silicate (MgSiO3) mezzanine characterized by uneven morphology and exposure of the metal substrate, while the surface of the coated plate is flat without prominent pores. There is a significant contrast from the morphology for judgment. Although we have carried out the EDS analysis, the results show that the elements in the blank plate are Si, O, Mg and Fe, while the coated plate is Si, O, Al, P and Mg, which can only prove the coverage of the coating and does not further promote the analysis of the mechanism. Therefore, we only present Figure 4 in the manuscript as the SEM results, without adding EDS results.
Point 9: Why the authors are not performed the XRD analysis to evaluate the formation of different phases on the substrate
Response 9: Thank you for your suggestion. In the process of designing the experimental methods, our initial phase observation strategy was to coat the same coating on the same batch of oriented silicon steel sheets, and conduct XRD analysis after different heat treatments. The results show that it is difficult to exclude the signal of the oriented silicon steel substrate even with the glancing incidence since the coating thickness is only 1-2 μm while the substrate is 270μm. The signal of the substrate is too strong to analyze the signal of the coating. We also try to scrape the coating from the coated plate. But it’s impossible to ensure that the MgSiO3 mezzanine will not be scraped off, since the hardness of the coating is high and the thickness is small. Finally, we choose to prepare the coating powder for XRD analysis.
Point 10: Authors are not measured the mass loss/gain after the samples are exposed to the environment
Response 10: Thank you for your suggestion. We did not present weight change monitoring in the manuscript for two reasons:
1. We have used an electrochemical test and NSS to evaluate the corrosion resistance. The electrochemical test can sensitively observe the micro electrochemical reaction at the interface between the coating and the substrate, and can effectively and rapidly evaluate the corrosion resistance of the coating. NSS is a standardized operation based on national and international standards, which provides a horizontal comparison possibility with high reliability in research and industrial application. In the NSS, the corrosion area counted is the final evaluation method, so we have recorded the corrosion area rather than the quality change.
2. The oriented silicon steel sheet is a sandwich structure. In the process of corrosion deepening and diffusion, there is not only the peeling of the coating but also the peeling of the MgSiO3 mezzanine. The description of this phenomenon will be more detailed in our other manuscripts. However, due to this phenomenon, the accuracy of the weight change monitoring for the oriented silicon steel during the corrosion process is less than that of other metal materials such as stainless steel and Q235 steel.
In the future research, we will look for more appropriate experimental strategies to achieve effective and accurate weight change monitoring.
Point 11: In the present study the authors performed the study in 3.5%NaCL environment. Is the samples are exposed in the 3.5%NaCL environment in the real time application
Response 11: Thank you for your suggestion. The use of 3.5% NaCl solution in the electrochemical analysis is based on reports about anti-corrosion coating and our experimental needs. The actual service environment of the oriented silicon steel is as core laminations in transformers, motors or generators. Most of the corrosion occurs in the storage and processing of steel and is the atmospheric corrosion. The 3.5% NaCl solution is similar to the marine environment. It is also the Cl- concentration region where the metal corrosion rate is faster according to reports([1] Wang, X.; Xiao, K.; Cheng, X.Q.; Dong, C.F.; Wu, J.S.; Yi, P.; Mao, C.L.; Jiang, L.; Li, X.G. Corrosion prediction mode of Q235 steel in polluted marine atmospheric environment. Journal of Materials Engineering 2017, 45, 51-57. https://doi.org/10.11868/j.issn.1001-4381.2015.001414. [2] Wang, H.T.; Han, E.H.; Ke, W. Gray model and gray relation analysis for atmospheric corrosion of carbon steel and low steel. Corrosion Science and Protection Technology 2006,18,278-280. https://doi.org/10.3969/j.issn.1002-6495.2006.04.012. ). And in the reports about anti-corrosion coating, the 3.5% NaCl solution is widely used as the electrolyte solution for electrochemical analysis to test the anti-corrosion performance of organic and semi-inorganic coatings ([3] Wang, S.H.; Long, Y.F.; Zhao, S.L.; An, C.Q.; Ding, K. A New Chromate-free insulating coating on Silicon Steel. Advanced Materials Research 2014, 1004–1005, 757–762. https://doi.org/10.4028/www.scientific.net/AMR.1004-1005.757. [4] Shi, C.; Shao, Y.; Wang, Y.; Meng, G.; Liu, B. Influence of submicro-sheet zinc phosphate modified by urea-formaldehyde on the corrosion protection of epoxy coating. Surfaces and Interfaces 2020, 18, 100403. https://doi.org/10.1016/j.surfin.2019.100403. [5] Wang, Y. N.; Dai, X. Y.; Xu, T. L.; Qu, L. J.; Zhang, C. L. Preparation and anticorrosion properties of saline grafted nano-silica/epoxy composite coating. Chemical Journal of Chinese Universities 2018, 39, 1564-1572. https://doi.org/10.7503/cjcu20170821.). The use of the same solution facilitates the performance comparison of our coatings with those reported. In addition, this concentration is lower than the NaCl concentration in NSS. This is because we consider that the solution in NSS is in the spraying state and does not stay on the plate surface, while the plate surface acts as the working electrode in the electrochemical test and needs to be immersed in the solution. In order to meet the needs of our other experimental strategies, we finally chose 3.5% NaCl as the electrochemical test environment.
Point 12: Why the corrosion mechanism are not derived in the present investigation
Response 12: Thank you for your suggestion. In this manuscript, we mainly present the anti-corrosion mechanism of the coating, namely the shielding effect. And we focus more on the influence of coating components adjustment, such as the change of silica sol particle size, on the shielding effect of the coating, that is, the action mechanism shown in Figure 8 in the manuscript. Through TGA, XRD and SEM analysis, it can be speculated that the structure of the coating is the combination of AlPO4 which provides structural stability through a three-dimensional network structure and stacked SiO2 particles. And it still has a complete structure to realize effective coverage after high-temperature annealing. For the coating based on the shielding effect, making the path of the corrosive more tortuous will be an effective way to improve corrosion resistance. Based on this analysis, we take the SiO2 as the example and summarize the effect of coating components on coating performance as shown in Figure 8 in the manuscript.
Figure 8. Scheme of the effect of silica sol particle size on the corrosion resistance
In the figure the film thickness is fixed. This is because the thinner the insulating coating is, the better. The coating thickness on the surface of oriented silicon steel is usually 0.5-2.5μm. Otherwise, the thick film would cause the decrease of metal proportion and influence the application of iron core which is consisted of oriented silicon steel sheets stacked up. In the figure it is also observed that some SiO2 particles are above the AlPO4 net and some particles are under the net. It’s based on the SEM and XRD analysis results and we believe that the Al(H2PO4)3 transforms into AlPO4 constituting a three-dimensional network structure and the silica sol transforms into SiO2 particles interspersing in the net and stacking on the substrate. The length of the corrosive diffusion path in the coating consisted of normal size SiO2 particles is L1. When the thickness is fixed, increasing the size would reduce the amount of SiO2 or even the number of layers and the intergranular pore width would increase. The length of the corrosive diffusion path in such coating is L2. On the contrary, decreasing the size would increase the amount of SiO2, reduce the intergranular pore width and change the length of the corrosive diffusion path to L3. When combining particles of different sizes, the small particles would fill the blank among big particles, the path would be more complex and irregular. The length in such coatings is L4. It can be found that L3 is greater than L1 and L1 is greater than L2. L4 is depended on the difference between the two particle sizes, and the larger the difference, the more complex the path will be, that is, the increase of tortuosity. The increase of the length and tortuosity of the corrosion penetration path can delay or even prevent the corrosion, and realize the increase of the corrosion resistance of the coating. The addition of MgO, Li2SiO3, K2SiO3, or LiOH is also believed to help make the path of the corrosive more tortuous. In addition, our experiments show that the micro electrochemical process at the shielding layer/metal interface has not been changed in our system. What happens at the interface is the dissolution of metal at the anode and the reduction of oxygen at the cathode as reported in the field of corrosion science. Of course, in our further research, we also found that the use of some functional fillers will have an impact on the micro electrochemical process at the shielding layer/metal interface. At that time, we will elaborate on the corrosion mechanism in more detail.
Point 13: Section title 3, should be rewrite as Results and Discussion
Response 13: Thank you for your suggestion. We correct the title of Section 3 to Results and Discussion and highlighted it in yellow.
Point 13: Section title 3, should be rewrite as Results and Discussion
Response 13: Thank you for your suggestion.We correct the title of Section 3 to Results and Discussion and highlighted in yellow.
Point 13: Section title 3, should be rewrite as Results and Discussion
Response 13: Thank you for your suggestion. We correct the title of Section 3 to Results and Discussion and highlighted in yellow.

Reviewer 3 Report
It is a good paper nicely presented proposing a new environmentally friendly coating for oriented silicon steel. Some minor revision need to be done. When revise your paper kindly have in mind the following:
2. The resolution of the following figures need to be enhanced- Figure 1,2,3,6 and 7.
3. In figure 1 the EXO- up or down- transformation must be indicated using text or arrow.
4. The authors claim that: -That indicates the phase structure of the coating is changing during the heat process.-will add value to the paper if there will be inserted the phase diagram indicating the specific phase transformation.
5. Discussing XRD plots the authors should consider the lattice parameters analyzing of XRD peak profiles-full-width at half-maximum (FWHM)- knowing that it is sensitive to the variation in microstructure and stress–strain accumulation in the layer and finally influencing adhesion on the substrate.
6. In the Reference chapter check once again the references format.
Author Response
Response to Reviewer 3 Comments
Point 1: The resolution of the following figures need to be enhanced- Figure 1,2,3,6 and 7.
Response 1: Thank you for your suggestion. We have updated the images in the manuscript, adjusted settings to avoid loss of clarity due to compression, and uploaded image source files to the submission system.
Point 2: In figure 1 the EXO- up or down- transformation must be indicated using text or arrow.
Response 2: Thank you for your suggestion. We have added arrows in Figure 1.
Point 3: The authors claim that: -That indicates the phase structure of the coating is changing during the heat process.-will add value to the paper if there will be inserted the phase diagram indicating the specific phase transformation.
Response 3: Thank you for your suggestion. Although the main components of our coating are aluminum phosphate (AlPO4) and silica (SiO2) after curing, the coating formulation also contains other metal oxides, which can affect the phase composition of the coating. And at the coating/magnesium silicate (MgSiO3) mezzanine interface, interactions exist. The phase inversion process cannot be presented by binary or ternary phase diagrams. In addition, due to the instrument limitations, it is still difficult to make a more precise distinction about the phase composition. Therefore, we will focus on the phase transformation process of the coating during the heat treatment process in the further study, and find a more suitable experimental strategy to realize the characterization of this process.
Point 4: Discussing XRD plots the authors should consider the lattice parameters analyzing of XRD peak profiles-full-width at half-maximum (FWHM)- knowing that it is sensitive to the variation in microstructure and stress–strain accumulation in the layer and finally influencing adhesion on the substrate.
Response 4: Thank you for your suggestion. We also agree that the lattice parameters, especially the FWHM, obtained by in-depth analysis of the XRD patterns are important for the calculation of grain size and stress-strain, and that the stress-strain is crucial for coating adhesion and magnetic properties of grain-oriented silicon steel. However, due to the instrument limitations, we found that it was very difficult to select relevant parameters during the fitting process, and it was difficult to obtain effective fitting results. Therefore, we did not obtain sufficient data to support the analysis by XRD. At present, we have tried to obtain the surface tension of the coating by measuring the static contact angle and obtaining the tension by the bending test to analyze the stress-strain change, but the core of this manuscript is to investigate the influence of the coating components on the corrosion resistance, so we didn’t mention this part in our manuscript. In the following research, we will combine the characteristics of the coating to find a more suitable experimental strategy to realize the characterization and analysis of the microstructure and stress-strain of the coating.
Point 5: In the Reference chapter check once again the references format.
Response 5: Thank you for your suggestion. We have checked the reference section.

Reviewer 4 Report
The work submitted by the authors is interesting and timely. Improving the functional performance of silicon steels from which the component parts of high-power transformers are made. The authors present a technology for improving silicon steel by surface coating with materials that they propose, experiment with, test and present the results.
The research carried out by the authors is interesting, topical, the results are presented explicitly, the diagrams, figures, tables are sufficient to support the conclusions of the authors.
The paper presents 34 bibliographical references, 27 references are used in part 1, Introduction, of the paper. Most of these bibliographic references are from recent years.
The similarity test gave a result of 12%, most of the similarities being reported when taking without citation some data presented in two other works, the first " Preparation of reduced graphene oxide and its application in chromium-free inorganic insulating coating for oriented silicon steel" autors Ying Liu, Lin Wu, Ling Tong, Xiaoyu Yang, Ao Chen, Yilai Zhou, Zhiyuan Liao, Baoguo Zhang and Ya Hu, published in Journal of Physics: Conference Series, Volume 2076, 7th International Conference on Energy Technology and Materials Science (ICETMS 2021) 27-29 September 2021, Zhoushan, China, paper presented in the bibliographical references, position 15, but from which sentences are taken without citation and the second" Three-Dimensional Acoustic Analysis of a Rectangular Duct with Gradient Cross-Sections in High-Speed Trains:A Theoretical Derivation", autors Yanhong Sun, Yi Qiu, Lianyun Liu and Xu Zheng published in Appl. Sci. 2022, 12, 5307. https://doi.org/10.3390/app12115307, work not mentioned in the bibliographical references of the article.
The value given by the similarity test was acceptable.

Author Response
Response to Reviewer 4 Comments
Point 1: The work submitted by the authors is interesting and timely. Improving the functional performance of silicon steels from which the component parts of high-power transformers are made. The authors present a technology for improving silicon steel by surface coating with materials that they propose, experiment with, test and present the results.
The research carried out by the authors is interesting, topical, the results are presented explicitly, the diagrams, figures, tables are sufficient to support the conclusions of the authors.
The paper presents 34 bibliographical references, 27 references are used in part 1, Introduction, of the paper. Most of these bibliographic references are from recent years.
The similarity test gave a result of 12%, most of the similarities being reported when taking without citation some data presented in two other works, the first " Preparation of reduced graphene oxide and its application in chromium-free inorganic insulating coating for oriented silicon steel" autors Ying Liu, Lin Wu, Ling Tong, Xiaoyu Yang, Ao Chen, Yilai Zhou, Zhiyuan Liao, Baoguo Zhang and Ya Hu, published in Journal of Physics: Conference Series, Volume 2076, 7th International Conference on Energy Technology and Materials Science (ICETMS 2021) 27-29 September 2021, Zhoushan, China, paper presented in the bibliographical references, position 15, but from which sentences are taken without citation and the second" Three-Dimensional Acoustic Analysis of a Rectangular Duct with Gradient Cross-Sections in High-Speed Trains:A Theoretical Derivation", autors Yanhong Sun, Yi Qiu, Lianyun Liu and Xu Zheng published in Appl. Sci. 2022, 12, 5307. https://doi.org/10.3390/app12115307, work not mentioned in the bibliographical references of the article.
The value given by the similarity test was acceptable.
Response 1:Thank you for your suggestion. We have revised the content of the article and cited relevant literature. The citation of reference 15 in this manuscript is to introduce the current application state of the nanomaterials in anti-corrosion coating research. The repetition may be due to that this article is our another idea about the improvement of corrosion resistance of chromium-free coatings for grain-oriented silicon steel. And this report is an optimization in another direction based on the chromium-free inorganic initial formulation in our research group, so there are similarities in some descriptions.

Round 2
Reviewer 1 Report
Authors responded well for my concerns. In my opinion, the manuscript can be considered for publication in Coatings.
Reviewer 2 Report
No comments